# Catalytic Enantioselective Synthesis of N-C Axially Chiral *N*-(2,6-Disubstituted-phenyl)sulfonamides through Chiral Pd-Catalyzed *N*-Allylation

**DOI:** 10.3390/molecules27227819

**Published:** 2022-11-13

**Authors:** Sota Fukasawa, Tatsuya Toyoda, Ryohei Kasahara, Chisato Nakamura, Yuuki Kikuchi, Akiko Hori, Gary J. Richards, Osamu Kitagawa

**Affiliations:** 1Department of Applied Chemistry (Japanese Association of Bio-Intelligence for Well-Being), Shibaura Institute of Technology, 3-7-5 Toyosu, Kohto-ku, Tokyo 135-8548, Japan; 2Graduate School of Engineering and Science, Shibaura Institute of Technology, 307 Fukasaku, Minuma-ku, Saitama 337-8570, Japan

**Keywords:** axial chirality, atropisomers, sulfonamides, palladium, *N*-allylation, asymmetric catalyst

## Abstract

Recently, catalytic enantioselective syntheses of N-C axially chiral compounds have been reported by many groups. Most N-C axially chiral compounds prepared through a catalytic asymmetric reaction possess carboxamide or nitrogen-containing aromatic heterocycle skeletons. On the other hand, although N-C axially chiral sulfonamide derivatives are known, their catalytic enantioselective synthesis is relatively underexplored. We found that the reaction (Tsuji–Trost allylation) of allyl acetate with secondary sulfonamides bearing a 2-arylethynyl-6-methylphenyl group on the nitrogen atom proceeds with good enantioselectivity (up to 92% ee) in the presence of (*S*,*S*)-Trost ligand-(allyl-PdCl)_2_ catalyst, affording rotationally stable N-C axially chiral *N*-allylated sulfonamides. Furthermore, the absolute stereochemistry of the major enantiomer was determined by X-ray single crystal structural analysis and the origin of the enantioselectivity was considered.

## 1. Introduction

Atropisomers (N-C axially chiral compounds), owing to the rotational restriction around an N-C single bond, have recently attracted much attention [1,2,3,4,5,6,7]. In 2002 and 2005, we reported the enantioselective syntheses of *ortho-tert*-butyl anilides **IA** and **IB** through chiral Pd-catalyzed *N*-allylation (Tsuji–Trost allylation) and *N*-arylation (Buchwald–Hartwig amination), respectively (Figure 1a) [8,9]. The *N*-allylation reaction shown in Figure 1a was the first example of the catalytic asymmetric synthesis of N-C axially chiral compounds [8], although the enantioselectivity was by no means satisfactory. The enantioselectivity was significantly improved by using *N*-arylation instead of *N*-allylation, and *N*-arylated anilide products **IB** were obtained in 88–96% ee [9]. In 2010, as the first catalytic asymmetric synthesis of non-amide type N-C axially chiral compounds, we succeeded in the enantioselective construction of *N*-(*ortho-tert*-butylphenyl)-2-arylindoles **II** through chiral Pd(II)-catalyzed 5-*endo*-hydroaminocyclization of 2-alkynyl aniline derivatives (Figure 1b) [10]. Since the publication of the reactions shown in Figure 1a,b, N-C axially chiral compounds have been widely accepted as new target molecules for catalytic asymmetric reactions, and more than 130 original papers on their catalytic enantioselective syntheses have been published to date [2,3,4,5,6,7]. Most N-C axially chiral compounds, which have been prepared through catalytic asymmetric reactions, are carboxamide derivatives such as **I** or nitrogen-containing aromatic heterocycles such as **II**.

On the other hand, although N-C axially chiral sulfonamides are also known [11,12,13,14], their catalytic asymmetric synthesis was not reported until recently. Since some N-C axially chiral sulfonamides are pharmaceutically attractive compounds, their catalytic asymmetric synthesis is meaningful from the viewpoint of not only synthetic organic chemistry, but also medicinal chemistry. In 2019, we and Zhao et al. independently reported the catalytic asymmetric synthesis of N-C axially chiral sulfonamides **IIIA** and **IIIB** through *N*-allylation with a chiral Pd catalyst and a chiral organic base, respectively (Figure 1c,d) [15,16]. The products in Figure 1c (our reaction) were *N*-(*ortho*-mono-*tert*-butylphenyl)sulfonamides **IIIA**, which are rotationally somewhat unstable, while the products in Figure 1d (Zhao’s reaction) were *N*-(2,6-disubstituted-phenyl)sulfonamides **IIIB**, which are rotationally relatively stable. Subsequently, other groups also succeeded in the catalytic enantioselective synthesis of *N*-(*ortho*-mono-*tert*-butylphenyl) and *N*-(2,6-disubstituted-phenyl)sulfonamides through similar or other asymmetric reactions [17,18,19,20,21,22]. We were curious about whether our method via chiral Pd-catalyzed *N*-allylation can also be applied to the enantioselective synthesis of *N*-(2,6-disubstituted-phenyl)sulfonamides.

In this article, we report the catalytic enantioselective synthesis of N-C axially chiral *N*-(2,6-disubstituted-phenyl)sulfonamides through the chiral Pd-catalyzed *N*-allylation of secondary sulfonamides (Figure 2). It was found that *N*-allylation with *N*-(2-arylethynyl-6-methylphenyl)sulfonamides proceeded with good enantioselectivity in the presence of (*S*,*S*)-Trost ligand-(allyl-PdCl)_2_ to give rotationally stable N-C axially chiral sulfonamides in a reasonable yield. Furthermore, the absolute stereochemistry of the major enantiomer was determined and the origin of the enantioselectivity was rationally explained.

## 2. Results and Discussion

### 2.1. Survey of 2,6-Disubstituents on N-Aryl Group

It is well known that in an allylation using a chiral π-allyl-Pd catalyst, the asymmetric induction on a nucleophile is more difficult than that on an allyl group because a nucleophile approaches from the opposite site of the Pd atom possessing a chiral ligand (Figure 2) [23,24,25,26]. Trost ligands are demonstrated to provide the effective chiral circumstance for a highly asymmetric induction on a prochiral nucleophile [27,28,29]. Indeed, in the Pd-catalyzed *N*-allylation with *N*-(*ortho-tert*-butylphenyl)sulfonamides (Figure 1c), the use of other chiral ligands other than Trost ligands caused a significant decrease in enantioselectivity [15,30,31,32]. Hence, we explored the *N*-allylation of secondary sulfonamides **1** bearing various 2,6-disubstituted-phenyl groups in the presence of Trost ligand-Pd catalysts (screening of 2,6-disubstituents, Table 1).

The reaction of allyl acetate with the anion species prepared from 4-tosyl amide **1a–i** and NaH (1 equiv) in THF was conducted for 5–27 h at −20 °C in the presence of (*S*,*S*)-Trost ligand (4.4 mol%) and (allyl-Pd-Cl)_2_ (2.2 mol%)_._ In the *N*-allylation of 2-*tert*-butyl-6-methylphenyl derivative **1a**, the chemical yield and enantioselectivity (58%, 10% ee) were significantly lowered in comparison with those (quant, 73% ee) of the *ortho*-mono-*tert*-butylphenyl derivative (Entry 1). The reaction of 2-iodo-6-phenyl derivative **1b** did not proceed smoothly to give *N*-allylation product **2b** with a poor yield (10%) and low enantioselectivity (43% ee, Entry 2). Although the reaction of 2-bromo-6-phenyl derivative **1c** gave the product **2c** with a high yield (97%), the enantioselectivity was low (38%, Entry 3). With 2-iodo-6-methyl derivative **1d** and 2-methyl-6-stylyl derivative **1e**, the products **2d** and **2e** were obtained with high yields (94% and 91%) and moderate enantioselectivity (65% ee, Entries 4 and 5). After further screening of the *ortho*-substituents, it was found that *N*-allylation with *ortho*-tolylethynyl derivatives **1f–i** gave relatively good results (82%- quant, 72–86% ee, Entries 6–9), In particular, with 2-methyl-6-tolylethynyl derivative **1i**, a maximum enantioselectivity (86% ee) was observed (Entry 9). Attempts were made to improve the enantioselectivity using other Trost ligands possessing a cyclohexyl skeleton. However, a decrease in the enantioselectivity or chemical yield was observed (Entries 10 and 11).

### 2.2. Survey of Alkynyl Substituents

Subsequently, under the same conditions, alkynyl substituents of *N*-(2-ethynyl-6-methylphenyl)-4-toluenesulfonylamide substrate, which gave the best result in Table 1, were explored (Table 2). Similar to 4-tolylethynyl derivative **1i**, the reaction with (4-methoxylphenyl)ethynyl and phenylethynyl derivatives **1j** and **1k** also gave *N*-allylated products **2j** and **2k** with high yields (98 and 92%) and good enantioselectivities (88 and 89% ee, Entries 2 and 3). On the other hand, in the reaction with trimethylsilylethynyl and hexynyl derivatives **1l** and **1m**, a considerable decrease in the enantioselectivity was observed. In these cases, the products **2l** and **2m** were obtained in 75 and 77% ee, respectively (Entries 4 and 5).

### 2.3. Survey of Sulfonyl Substituents

The substituent effect on the sulfonyl group was further explored by using *N*-(2-arylethynyl-6-methylphenyl)sulfonamide substrates (Table 3).

The present reactions proceeded smoothly regardless of the electronic effect of the *para*-substituent on the benzenesulfonyl group, affording *N*-allylation products **2n**–**q** with high yields (88%–quant) and good enantioselectivities (85–92% ee, Entries 2–5). With benzenesulfonyl amides **1o,p** bearing an electron-withdrawing substituent such as a nitro group, a slight increase in enantioselectivity was observed (89 and 92% ee, Entries 3 and 4). The reaction of methanesulfonyl amides **1r** also gave the product **2r** with a good enantioselectivity (87% ee, Entry 6). On the other hand, in the reaction with bulky 2,4,6-trimethylphenylsulfone amide **1s**, the enantioselectivity was considerably lowered (63% ee, Entry 7).

### 2.4. Absolute Stereochemistry and Origin of Enantioselectivity

The absolute stereochemistry of the major enantiomer was determined to be (*P*)-configuration by X-ray single crystal structural analysis of **2o** (Figure 1) with the flack parameter 0.02(6) [33,34]. Although the absolute stereochemistries of other *ortho*-ethynyl sulfonamides **2f–s** were not determined exactly, the major enantiomers of **2f–s** (+61.5–196.7°), which have large positive [α]_D_ values such as **2o** (+201°), were also predicted to possess the (*P*)-configuration (only methanesulfonamide **2r** showed a small positive [α]_D_ value = +7.7°). Moreover, in the previously reported reaction of *N*-(*ortho*-mono-*tert*-butylphenyl)sulfonamides using (*S*,*S*)-Trost ligand (Figure 1c), since the *N*-allylated products **IIIA** possessing (*P*)-configuration were obtained as the major enantiomer, the ethynyl group is expected to act as a bulky substituent in a similar way to the *tert*-butyl group.

The (*P*)-selectivity in the present reaction may be rationalized on the basis of a working model proposed by Trost (Figure 2) [35,36]. Among four possible transition states **TS-A**–**D** in the reaction with (*S*,*S*)-Trost ligand, **TS-B** and **TS-C** should be significantly destabilized because of the strong steric repulsion between the *ortho*-ethynyl or *ortho*-methyl group and Ph (wall) group (green color) on the phosphorus atom. **TS-D** may also not be favorable, due to the steric repulsion between the *ortho*-ethynyl group and Ph (wall) group (blue color). As a result, the reaction preferentially proceeds via **TS-A**, leading to (*P*)-**2** as a major enantiomer. In other 2,6-disubstituted phenyl derivatives **1a-e** except for *ortho*-ethynyl derivatives, the reaction may proceed via **TS-D** as well as **TS-A**, resulting in the decrease in the enantioselectivity. Since a linear *ortho*-arylethynyl group brings about the considerable steric interaction with Ph (wall) groups (blue color) on the back side in **TS-D**, the reaction via **TS-D** may be disfavored, resulting in a good enantioselectivity. With a substrate **1s** bearing a bulky sulfonyl group (R^1^ = 2,4,6-Me_3_C_4_H_2_), the destabilization in **TS-A** may be caused by the steric repulsion between the Ph (wall) group on the front side and R^1^ substituent, leading to the decrease in the enantioselectivity (Table 3, Entry 7).

### 2.5. Rotational Stability of Sulfonamide Products

The rotational barriers of *N*-(*ortho*-mono-*tert*-butylphenyl)sulfonamide derivatives **IIIAr** and **IIIAs**, which were previously reported, were 25.2 and 25.5 kcal mol^−1^ at 298 K, respectively (Figure 3), and the ee of **IIIAr** and **IIIAs** decreased gradually at rt in CCl_4_ (*t*_1/2_ at 298K = 1.9 and 3.6 days). On the other hand, in *N*-(2-arylethynyl-6-methylphenyl)sulfonamide products **2r** and **2i**, the decrease in the ee was not observed even after standing for a few days at rt in CCl_4_. The barrier values of **2r** and **2i** were evaluated to be 28.3 and 28.7 kcal mol^−1^ at 333 K, which are ca. 3 kcal mol^−1^ higher than those of **IIIAr** and **IIIAs**.

In *N*-allyl-*N*-(2-(4-tolyl)ethynyl)phenyl sulfonamide **3** bearing no methyl group at the other *ortho*-position, the enantiomers could not be separated through a chiral HPLC method because of the rotationally unstable structure. Indeed, two allylic hydrogens (Ha and Hb) in **III** and **2** were detected as nonequivalent signals in the ^1^H NMR, while those in **3** showed an equivalent NMR signal, which suggests the quick rotation around the N-Ar bond at the NMR time scale (Appendix A). 

### 2.6. Application to Enantioselective Double N-Allylation

Since *N*-allyl-*N*-(2,6-disubstituted-phenyl)sulfonamide products **2** were revealed to be rotationally stable at rt, we further investigated the enantioselective construction of two N-C chiral axes through a double *N*-allylation with bis-sulfonamide substrate (Figure 3). In the presence of an achiral Pd catalyst, the double *N*-allylation with *N*-(2-bromo-6-tolylethynylphenyl)bis-sulfonamide **4** proceeds smoothly to give a 1:1 mixture of diastereomeric double allylation products *chiral*-**5** and *meso*-**5** (82% yield). The stereochemistry of both diastereomers was determined by chiral HPLC method. That is, the HPLC of one diastereomer (*chiral*-**5**) using a CHIRALPAK AD-H column gave two peaks corresponding to enantiomers, while for the other diastereomer (*meso*-**5**), the enantiomer separation by chiral HPLC was not observed. No isomerization between *chiral*-**5** and *meso*-**5** was detected even after standing for a several days at rt.

Subsequently, the enantioselective double *N*-allylation with **4** was conducted at −20 °C in the presence of (*S*,*S*)-Trost ligand-Pd catalyst. In this case, the double *N*-allylated products *chiral*-**5** and *meso*-**5** were obtained in a diastereomer ratio of 3.1:1 (88% yield). After the removal of *meso*-**5** via MPLC separation, the optical purity of the obtained *chira*l-**5** was found to be 99% ee. Since no the diastereoselectivity was observed at all under the achiral reaction conditions, it is obvious that the chiral axis constructed in the first *N*-allylation does not influence asymmetric induction in the second *N*-allylation (the stereoselectivity is only determined by the chiral catalyst).

The significantly high optical purity of double *N*-allylation product *chira*l-**5** in comparison with mono-*N*-allylation products **2** (for example, **2g**: 72% ee, Entry 7 in Table 1) can be rationally explained on the basis of the Horeau principle [37,38,39]. The product distributions (the enantiomeric excess and diastereomer ratio) in double asymmetric reactions are represented in Equations A and B (Figure 4). When the ee (72% ee, x = 0.86) of 2-bromo-6-arylethynyl derivative **2g** is used as the value (x) for the first asymmetric induction in Figure 3, the ee of *chiral*-**5** and the diastereomer ratio were calculated to be 95% and 3.2, respectively, which are similar to the experimental values (99% ee and dr = 3.1). Thus, it was revealed that bis-sulfonamide bearing two N-C chiral axes is obtained in a high optical purity through an asymmetric double *N*-allylation.

## 3. Materials and Methods

### 3.1. General Information

Melting points were uncorrected. ^1^H and ^13^C NMR spectra were recorded on a 400 MHz spectrometer. In ^1^H and ^13^C NMR spectra, chemical shifts were expressed in *δ* (ppm) downfield from CHCl_3_ (7.26 ppm) and CDCl_3_ (77.0 ppm), respectively. HRMS were recorded on a double-focusing magnetic sector mass spectrometer using electron impact ionization. Column chromatography was performed on silica gel (75–150 μm). Medium-pressure liquid chromatography (MPLC) was performed on a 25 × 4 cm i.d. prepacked column (silica gel, 10 μm) with a UV detector. High-performance liquid chromatography (HPLC) was performed on a 25 × 0.4 cm i.d. chiral column with a UV detector. Optical rotations were measured in CHCl_3_ or MeOH on JASCO P-1020 Polarimeter at λ = 589 nm. [α]_D_ values are reported at 25 °C in degree·cm^2^·g^–1^ with concentrations reported in g/100 mL.

### 3.2. Synthesis of Substrates 1 and Asymmetric N-Allylation with 1

*N*-(2-*tert*-Butyl-6-methylphenyl)-4-methylbenzenesulfonamide (**1a**). Under N_2_ atmosphere, to 2-*tert*-butyl-6-methylaniline (488 mg, 3.0 mmol, commercially available) and pyridine (0.36 mL, 4.5 mmol) in CH_2_Cl_2_ (4.0 mL) was added 4-tosyl chloride (631 mg, 3.3 mmol), and then the mixture was stirred for 22 h at 0 °C–rt. The mixture was poured into water and extracted with AcOEt. The AcOEt extracts were washed with brine, dried over MgSO_4_, and evaporated to dryness. Hexane was added to the residue and the mixture was filtered in vacuo. After washing the residue with hexane, **1a** was obtained (417 mg, 44%). **1a**: white solid; mp 157–160 °C; IR (neat) 3268, 1323, 1155 cm^−1^; ^1^H NMR (400 MHz, CDCl_3_) *δ*: 7.68 (2H, d, *J* = 8.1 Hz), 7.34 (1H, d, *J* = 8.1 Hz), 7.27 (2H, d, *J* = 8.1 Hz), 7.14 (1H, t, *J* = 7.6 Hz), 7.00 (1H, d, *J* = 7.1 Hz), 6.30 (1H, s), 2.43 (3H, s), 1.91 (3H, s), 1.43 (9H, s); ^13^C{^1^H} NMR (100 MHz, CDCl_3_) *δ*: 148.6, 143.2, 138.9, 138.6, 132.3, 129.4, 129.0, 127.4, 127.0, 126.4, 36.1, 32.4, 21.5, 20.1; MS (ESI-TOF) *m*/*z*: [M + Na]^+^ 340; HRMS (ESI-TOF) *m*/*z*: [M + Na]^+^ Calcd for C_18_H_23_NNaO_2_S 340.1347; Found 340.1347.

*N*-((2-Iodo-4-methyl-6-phenyl)phenyl)-4-methylbenzenesulfonamide (**1b**). Under N_2_ atmosphere, to phenylboronic acid (438 mg, 3.6 mmol) and potassium carbonate (1.66 g, 12.0 mmol) in H_2_O (10 mL) were added bis(triphenylphosphine)palladium(II) chloride (107 mg, 0.15 mmol) and 2,6-diiodo-4-methylaniline (1.04 g, 2.9 mmol). The mixture was stirred for 3 h at rt. The mixture was poured into water and extracted with AcOEt. The AcOEt extracts were washed with brine, dried over MgSO_4_, and evaporated to dryness. Purification of the residue by column chromatography (hexane/AcOEt = 150) gave 2-iodo-4-methyl-6-phenyl aniline (358 mg, 40%). Under N_2_ atmosphere, to 2-iodo-4-methyl-6-phenylaniline (610 mg, 2.0 mmol) and pyridine (0.24 mL, 3.0 mmol) in CH_2_Cl_2_ (6.0 mL) was added 4-tosyl chloride (414 mg, 2.2 mmol), and then the mixture was stirred for 22 h at 0 °C–rt. The mixture was poured into water and extracted with AcOEt. The AcOEt extracts were washed with brine, dried over MgSO_4_, and evaporated to dryness. Purification of the residue by column chromatography (hexane/AcOEt = 120 and then 5) gave **1b** (428 mg, 47%). **1b**: white solid; mp 135–136 °C; IR (neat) 3258, 1331, 1155 cm^−1^; ^1^H NMR (400 MHz, CDCl_3_) *δ*: 7.67 (1H, m), 7.26 (2H, dt, *J* = 8.5, 1.9 Hz), 7.19–7.22 (3H, m), 7.14–7.17 (2H, m), 7.06 (1H, d, *J* = 1.4 Hz), 7.01 (2H, d, *J* = 7.6 Hz), 6.42 (1H, s), 2.37 (3H, s), 2.31 (3H, s); ^13^C{^1^H} NMR (100 MHz, CDCl_3_) *δ*: 143.0, 141.8, 139.6, 139.5, 137.2, 132.6, 132.0, 129.2, 128.8, 128.0, 127.2, 127.0, 102.1, 21.5, 20.4; MS (ESI-TOF) *m*/*z*: [M + Na]^+^ 486; HRMS (ESI-TOF) *m*/*z*: [M + Na]^+^ Calcd for C_20_H_18_INNaO_2_S 486.0001; Found 485.9972.

*N*-((2-Bromo-4-methyl-6-phenyl)phenyl)-4-methylbenzenesulfonamide (**1c**). In accordance with the experimental procedure for the synthesis of **1b**, **1c** was prepared from 2-bromo-4-methyl-6-phenylaniline (318 mg, 1.2 mmol, commercially available) and 4-tosyl chloride (280 mg, 1.4 mmol). The reaction was conducted for 18 h at 0 °C–rt. Purification of the residue by column chromatography (hexane/AcOEt = 50 and then 3) gave **1c** (213 mg, 42%). **1c**: white solid; mp 149–152 °C; IR (neat) 3250, 1337, 1163 cm^−1^; ^1^H NMR (400 MHz, CDCl_3_) *δ*: 7.38 (1H, dd, *J* = 1.9, 1.0 Hz), 7.29 (2H, dt, *J* = 8.4, 1.9 Hz), 7.21–7.24 (5H, m), 7.06 (1H, d, *J* = 1.4 Hz), 7.03 (2H, d, *J* = 8.1 Hz), 6.42 (1H, s), 2.37 (3H, s), 2.33 (3H, s); ^13^C{^1^H} NMR (100 MHz, CDCl_3_) *δ*: 143.1, 142.5, 139.3, 139.2, 137.0, 132.7, 131.7, 129.2, 129.04, 128.97, 128.0, 127.1, 124.6, 21.5, 20.7; MS (ESI-TOF) *m*/*z*: [M + Na]^+^ 440; HRMS (ESI-TOF) *m*/*z*: [M + Na]^+^ Calcd for C_20_H_18_^81^BrNNaO_2_S 440.0119; Found 440.0102.

*N*-(2-Iodo-4,6-dimethylphenyl)-4-methylbenzenesulfonamide (**1d**). In accordance with the experimental procedure for the synthesis of **1b**, **1d** was prepared from 2-iodo-4,6-dimethylaniline (494 mg, 2.0 mmol, commercially available) and 4-tosyl chloride (419 mg, 2.2 mmol). The reaction was conducted for 17 h at 0 °C–rt. Purification of the residue by column chromatography (hexane/AcOEt = 10 and then 3) gave **1d** (804 mg, quant). **1d**: White solid; mp 161–163 °C; IR (neat) 3275, 1331, 1157 cm^−1^; ^1^H NMR (400 MHz, CDCl_3_) *δ*: 7.56 (2H, dt, *J* = 8.1, 1.9 Hz), 7.37 (1H, d, *J* = 1.4 Hz), 7.23 (2H, d, *J* = 8.1 Hz), 7.04 (1H, d, *J* = 1.4 Hz), 6.14 (1H, s), 2.45 (3H, s), 2.42 (3H, s), 2.24 (3H, s); ^13^C{^1^H} NMR (100 MHz, CDCl_3_) *δ*: 143.9, 139.4, 139.3, 137.5, 137.2, 133.2, 132.7, 129.5, 127.9, 100.2, 21.6, 20.7, 20.3; MS (ESI-TOF) *m*/*z*: [M + Na]^+^ 424; HRMS (ESI-TOF) *m*/*z*: [M + Na]^+^ Calcd for C_15_H_16_^127^INNaO_2_S 423.9844; Found 423.9816.

(*E*)-*N*-(2,4-Dimethyl-6-styrylphenyl)-4-methylbenzenesulfonamide (**1e**). In accordance with the experimental procedure for the synthesis of **1b**, **1e** was prepared from 2,4-dimethyl-6-styrylaniline (218 mg, 1.0 mmol) [40] and 4-tosyl chloride (279 mg, 1.5 mmol). The reaction was conducted for 22 h at 0 °C–rt. Purification of the residue by column chromatography (hexane/AcOEt = 5) gave **1e** (323 mg, 88%). **1e**: white solid; mp 150–152 °C; IR (neat) 3242, 1325, 1157 cm^−1^; ^1^H NMR (400 MHz, CDCl_3_) *δ*: 7.61 (2H, dt, *J* = 8.1, 1.9 Hz), 7.21–7.30 (4H, m), 7.16 (2H, dd, *J* = 8.1, 1.9 Hz), 7.06 (2H, d, *J* = 7.6 Hz), 6.99 (1H, d, *J* = 1.9 Hz), 6.83 (1H, d, *J* = 16.6 Hz), 6.74 (1H, d, *J* = 16.6 Hz), 6.30 (1H, s), 2.33 (3H, s), 2.30 (3H, s), 2.18 (3H, s); ^13^C{^1^H} NMR (100 MHz, CDCl_3_) *δ*: 143.6, 138.8, 137.9, 137.4, 137.0, 136.3, 131.3, 129.9, 129.6, 129.2, 128.3, 127.6, 127.1, 126.6, 124.3, 124.2, 21.3, 21.1, 19.0; MS (ESI-TOF) *m*/*z*: [M + Na]^+^ 400; HRMS (ESI-TOF) *m*/*z*: [M + Na]^+^ Calcd for C_23_H_23_NNaO_2_S 400.1347; Found 400.1330.

*N*-(2-Iodo-4-methyl-6-(*p*-tolylethynyl)phenyl])-4-methylbenzenesulfonamide (**1f**). Under N_2_ atmosphere, to 2,6-diiodo-4-methylaniline (1.08 g, 3.0 mmol, commercially available), copper iodide(I) (11.4 mg, 0.060 mol) and bis(triphenylphosphine)palladium(II)dichloride (42 mg, 0.060 mmol) in triethylamine (15 mL) was added 4-ethynyltoluene (384 mg, 3.3 mmol), and then the mixture was stirred for 19 h at rt. The mixture was poured into 2N HCl aqueous solution and extracted with AcOEt. The AcOEt extracts were washed with brine, dried over MgSO_4_, and evaporated to dryness. Purification of residue by column chromatography (hexane/AcOEt = 200) gave 2-iodo-4-methyl-6-(4-tolylethynyl)aniline (454 mg, 44%). In accordance with the experimental procedure for the synthesis of **1b**, **1f** was prepared from 2-iodo-4-methyl-6-(4-tolylethynyl)aniline (355 mg, 1.0 mmol) and 4-tosyl chloride (211 mg, 1.1 mmol). The reaction was conducted for 24 h at 0 °C–rt. Purification of the residue by column chromatography (hexane/AcOEt = 20 and then 5) gave **1f** (181 mg, 35%). **1f**: White solid; mp 208–211 °C; IR (neat) 3231, 2216, 1337, 1165 cm^−1^; ^1^H NMR (400 MHz, CDCl_3_) *δ*: 7.66 (1H, d, *J* = 1.4 Hz), 7.62 (2H, dt, *J* = 8.1, 1.9 Hz), 7.22–7.28 (3H, m), 7.12 (2H, d, *J* = 8.1 Hz), 7.08 (2H, d, *J* = 8.1 Hz), 6.49 (1H, s), 2.37 (3H, s), 2.28 (3H, s), 2.25 (3H, s); ^13^C{^1^H} NMR (100 MHz, CDCl_3_) *δ*: 143.6, 140.7, 139.1, 138.9, 137.6, 135.9, 133.6, 131.6, 129.5, 128.9, 127.6, 123.5, 119.3, 99.8, 94.8, 85.0, 21.6, 21.5, 20.3; MS (ESI-TOF) *m*/*z*: [M + Na]^+^ 524; HRMS (ESI-TOF) *m*/*z*: [M + Na]^+^ Calcd for C_23_H_20_^127^INNaO_2_S 524.0157; Found 524.0130.

*N*-(2-Bromo-4-methyl-6-(*p*-tolylethynyl)phenyl)-4-methylbenzenesulfonamide (**1g**). Under N_2_ atmosphere, to 2-bromo-4-methyl-6-iodoaniline (469 mg, 1.5 mmol, commercially available), copper iodide(I) (5.7 mg, 0.030 mmol) and bis(triphenylphosphine)palladium(II) dichloride (21 mg 0.030 mmol) in triethylamine (7.5 mL) was added 4-ethynyltoluene (192 mg, 1.7 mmol), and then the mixture was stirred for 22 h at rt. The mixture was poured into 2N HCl aqueous solution and extracted with AcOEt. The AcOEt extracts were washed with brine, dried over MgSO_4_, and evaporated to dryness. Purification of residue by column chromatography (hexane/AcOEt = 30) gave 2-bromo-4-methyl-6-(4-tolylethynyl)aniline (353 mg, 78%). In accordance with the experimental procedure for the synthesis of **1b**, **1g** was prepared from 2-bromo-4-methyl-6-(4-tolylethynyl)aniline (903 mg, 3.0 mmol) and 4-tosyl chloride (636 mg, 3.3 mmol). The reaction was conducted for 24 h at 0 °C–rt. Purification of the residue by column chromatography (hexane/AcOEt = 20 and then 3) gave **1g** (618 mg, 45%). **1g**: white solid; mp 191–197 °C; IR (neat) 3229, 2218, 1339, 1165 cm^−1^; ^1^H NMR (400 MHz, CDCl_3_) *δ*: 7.65 (2H, dt, *J* = 8.5, 1.9 Hz), 7.37 (1H, d, *J* = 1.4 Hz), 7.26–7.28 (3H, m), 7.09–7.13 (4H, m), 6.43 (1H, s), 2.37 (3H, s), 2.30 (3H, s), 2.27 (3H, s); ^13^C{^1^H} NMR (100 MHz, CDCl_3_) *δ*: 143.6, 138.92, 138.85, 137.6, 134.0, 132.8, 132.7, 131.7, 129.5, 128.9, 127.5, 124.5, 123.5, 119.4, 95.0, 85.0, 21.6, 21.5, 20.6; MS (ESI-TOF) *m*/*z*: [M + Na]^+^ 478; HRMS (ESI-TOF) *m*/*z*: [M + Na]^+^ Calcd for C_23_H_20_^81^BrNNaO_2_S 478.0275; Found 478.0249.

*N*-(2-Chloro-6-(*p*-tolylethynyl)phenyl)-4-methylbenzenesulfonamide (**1h**). Under N_2_ atmosphere, to 2-chloro-6-iodoaniline (507 mg, 2.0 mmol, commercially available), copper iodide(I) (7.5 mg, 0.040 mmol) and bis(triphenylphosphine)palladium(II) dichloride (29 mg, 0.041 mol) in triethylamine (10 mL) was added 4-ethynyltoluene (255 mg, 2.2 mmol), and then the mixture was stirred for 19 h at rt. The mixture was poured into 2N HCl aqueous solution and extracted with AcOEt. The AcOEt extracts were washed with brine, dried over MgSO_4_, and evaporated to dryness. Purification of residue by column chromatography (hexane/AcOEt = 30) gave 2-chloro-6-(4-tolylethynyl)aniline (398 mg, 82%). In accordance with the experimental procedure for the synthesis of **1b**, **1h** was prepared from 2-chloro-6-(4-tolylethynyl)aniline (214 mg, 0.9 mmol) and 4-tosyl chloride (189 mg, 1.0 mmol). The reaction was conducted for 24 h at 0 °C–rt. Purification of the residue by column chromatography (hexane/AcOEt = 30 and then 5) gave **1h** (121 mg, 34%). **1h**: white solid; mp 192–193 °C; IR (neat) 3229, 2209, 1337, 1165 cm^−1^; ^1^H NMR (400 MHz, CDCl_3_) *δ*: 7.67 (2H, d, *J* = 8.1 Hz), 7.40 (1H, dd, *J* = 7.6, 1.4 Hz), 7.36 (1H, dd, *J* = 8.1, 1.4 Hz), 7.30 (2H, d, *J* = 8.1 Hz), 7.10–7.19 (5H, m), 6.52 (1H, s), 2.38 (3H, s), 2.29 (3H, s); ^13^C{^1^H} NMR (100 MHz, CDCl_3_) *δ*: 143.7, 139.1, 137.4, 134.2, 133.3, 131.7, 131.3, 130.2, 129.5, 129.0, 128.0, 127.4, 124.7, 119.2, 95.7, 84.5, 21.6, 21.5; MS (ESI-TOF) *m*/*z*: [M + Na]^+^ 418; HRMS (ESI-TOF) *m*/*z*: [M + Na]^+^ Calcd for C_22_H_18_^35^ClNNaO_2_S 418.0645; Found 418.0615.

*N*-(2,4-Dimethyl-6-(*p*-tolylethynyl)phenyl)-4-methylbenzenesulfonamide (**1i**). Under N_2_ atmosphere, to 2,4-dimethyl-6-iodoaniline (494 mg, 2.0 mmol, commercially available), copper iodide(I) (7.6 mg, 0.040 mmol) and bis(triphenylphosphine)palladium(II) dichloride (28 mg, 0.040 mmol) in triethylamine (10 mL) was added 4-ethynyltoluene (254 mg, 2.2 mmol), and then the mixture was stirred for 19 h at rt. The mixture was poured into 2N HCl aqueous solution and extracted with AcOEt. The AcOEt extracts were washed with brine, dried over MgSO_4_, and evaporated to dryness. Purification of residue by column chromatography (hexane/AcOEt = 15) gave 2-bromo-4-methyl-6-(4-tolylethynyl)aniline (468 mg, 99%). In accordance with the experimental procedure for the synthesis of **1b**, **1i** was prepared from 2,4-dimethyl-6-(4-tolylethynyl)aniline (708 mg, 3.0 mmol) and 4-tosyl chloride (629 mg, 3.3 mmol). The reaction was conducted for 27 h at 0 °C–rt. Purification of the residue by column chromatography (hexane/AcOEt = 15 and then 5) gave **1i** (918 mg, 78%). **1i**: white solid; mp 149–152 °C; IR (neat) 3248, 2205, 1331, 1163 cm^−1^; ^1^H NMR (400 MHz, CDCl_3_) *δ*: 7.50 (2H, dt, *J* = 8.5, 1.9 Hz), 7.11–7.16 (4H, m), 7.06 (1H, s), 7.03 (1H, s), 6.99 (2H, d, *J* = 8.1), 6.44 (1H, s), 2.51 (3H, s), 2.38 (3H, s), 2.27 (3H, s), 2.23 (3H, s); ^13^C{^1^H} NMR (100 MHz, CDCl_3_) *δ*: 143.3, 138.6, 137.9, 136.9, 136.5, 132.7, 132.3, 131.3, 130.4, 129.3, 128.9, 127.4, 121.5, 119.4, 94.0, 84.8, 21.5, 21.4, 20.7, 19.4; MS (ESI-TOF) *m*/*z*: [M + Na]^+^ 412; HRMS (ESI-TOF) *m*/*z*: [M + Na]^+^ Calcd for C_24_H_23_NNaO_2_S 412.1347; Found 412.1332.

*N*-(2-((4-Methoxyphenyl)ethynyl)-4,6-dimethylphenyl)-4-methylbenzenesulfonamide (**1j**). Under N_2_ atmosphere, to 2,4-dimethyl-6-iodoaniline (619 mg, 2.5 mmol, commercially available), copper iodide(I) (10 mg, 0.053 mmol) and bis(triphenylphosphine)palladium(II) dichloride (35 mg, 0.050 mmol) in triethylamine (10 mL) was added 4-ethynylanisole (363 mg, 2.7 mmol), and then the mixture was stirred for 17 h at rt. The mixture was poured into 2N HCl aqueous solution and extracted with AcOEt. The AcOEt extracts were washed with brine, dried over MgSO_4_, and evaporated to dryness. Purification of residue by column chromatography (hexane/AcOEt = 30) gave 2,4-dimerthyl-6-((4-tmethoxyphenyl)ethynyl)aniline (402 mg, 64%). In accordance with the experimental procedure for the synthesis of **1b**, **1j** was prepared from 2,4-dimethyl-6-((4-methoxyphenyl)ethynyl)aniline (402 mg, 1.6 mmol) and 4-tosyl chloride (336 mg, 1.8 mmol). The reaction was conducted for 19 h at 0 °C–rt. Purification of the residue by column chromatography (hexane/AcOEt = 10 and then 5) gave **1j** (384 mg, 59%). **1j**: white solid; mp 149–150 °C; IR (neat) 3264, 2203, 1329, 1157 cm^−1^; ^1^H NMR (400 MHz, CDCl_3_) *δ*: 7.50 (2H, dt, *J* = 8.5, 1.9 Hz), 7.19 (2H, dt, *J* = 9.0, 2.4 Hz), 7.05 (1H, s), 7.01 (1H, s), 7.00 (2H, d, *J* = 7.6 Hz), 6.84 (2H, dt, *J* = 9.0, 2.4 Hz), 6.41 (1H, s), 3.84 (3H, s), 2.49 (3H, s), 2.27 (3H, s), 2.25 (3H, s); ^13^C{^1^H} NMR (100 MHz, CDCl_3_) *δ*: 159.7, 143.4, 137.9, 136.9, 136.6, 132.9, 132.6, 132.3, 130.3, 129.3, 127.5, 121.6, 114.7, 113.8, 93.9, 84.2, 55.3, 21.5, 20.7, 19.5; MS (ESI-TOF) *m*/*z*: [M + Na]^+^ 428; HRMS (ESI-TOF) *m*/*z*: [M + Na]^+^ Calcd for C_24_H_23_NNaO_3_S 428.1296; Found 428.1282.

*N*-(2,4-Dimethyl-6-(phenylethynyl)phenyl)-4-methylbenzenesulfonamide (**1k**). In accordance with the experimental procedure for the synthesis of **1b**, **1k** was prepared from 2,4-dimethyl-6-(4-phenyl)ethynyl)aniline (379 mg, 1.7 mmol, commercially available) and 4-tosyl chloride (357 mg, 1.9 mmol). The reaction was conducted for 22 h at 0 °C–rt. Purification of the residue by column chromatography (hexane/AcOEt = 20 and then 10) gave racemic **1k** (582 mg, 90%). white solid; mp 186–188 °C; IR (neat) 3241, 2212, 1331, 1165 cm^−1^; ^1^H NMR (400 MHz, CDCl_3_) *δ*: 7.51 (2H, dt, *J* = 8.1, 2.4 Hz), 7.24–7.34 (5H, m), 7.08 (1H, s), 7.05 (1H, s), 6.98 (2H, d, *J* = 7.6 Hz), 6.42 (1H, s), 2.51 (3H, s), 2.28 (3H, s), 2.22 (3H, s); ^13^C{^1^H} NMR (100 MHz, CDCl_3_) *δ*: 143.4, 138.0, 137.0, 136.6, 132.9, 132.4, 131.4, 130.5, 129.3, 128.5, 128.1, 127.4, 122.6, 121.3, 93.7, 85.5, 21.4, 20.7, 19.4; MS (ESI-TOF) *m*/*z*: [M + Na]^+^ 398; HRMS (ESI-TOF) *m*/*z*: [M + Na]^+^ Calcd for C_23_H_21_NNaO_2_S 398.1191; Found 398.1179.

*N*-(2,4-Dimethyl-6-((trimethylsilyl)ethynyl)phenyl)-4-methylbenzenesulfonamide (**1l**). In accordance with the experimental procedure for the synthesis of **1b**, **1l** was prepared from 2,4-dimethyl-6-(4-trimethylsilyl)ethynyl)aniline (434 mg, 2.0 mmol) [41] and 4-tosyl chloride (419 mg, 2.2 mmol). The reaction was conducted for 22 h at 0 °C–rt. Purification of the residue by column chromatography (hexane/AcOEt = 20 and then 5) gave **1l** (587 mg, 79%). **1l**: white solid; mp 112–113 °C; IR (neat) 3225, 2158, 1335, 1165 cm^−1^; ^1^H NMR (400 MHz, CDCl_3_) *δ*: 7.47 (2H, dt, *J* = 8.5, 1.9 Hz), 7.17 (2H, d, *J* = 8.5 Hz), 7.05 (1H, s), 6.98 (1H, s), 6.37 (1H, s), 2.47 (3H, s), 2.40 (3H, s), 2.24 (3H, s), 0.13 (9H, m); ^13^C{^1^H} NMR (100 MHz, CDCl_3_) *δ*: 143.4, 137.3, 136.6, 136.4, 133.2, 132.8, 130.7, 129.3, 127.8, 120.9, 100.6, 99.6, 21.6, 20.7, 19.5, −0.17; MS (ESI-TOF) *m*/*z*: [M + Na]^+^ 394; HRMS (ESI-TOF) *m*/*z*: [M + Na]^+^ Calcd for C_20_H_25_NNaO_2_S^28^Si 394.1273; Found 394.1257.

*N*-(2-(Hex-1-yn-1-yl)-4,6-dimethylphenyl)-4-methylbenzenesulfonamide (**1m**). In accordance with the experimental procedure for the synthesis of **1b**, **1m** was prepared from 2,4-dimethyl-6-(2-hex-1-yn-1-yl)aniline (363 mg, 1.8 mmol, commercially available) and 4-tosyl chloride (378 mg, 2.0 mmol). The reaction was conducted for 26 h at 0 °C–rt. Purification of the residue by column chromatography (hexane/AcOEt = 50 and then 30) gave **1m** (498 mg, 78%). **1m**: orange solid; mp 71–73 °C; IR (neat) 3233, 2228, 1333, 1165 cm^−1^; ^1^H NMR (400 MHz, CDCl_3_) *δ*: 7.50 (2H, d, *J* = 8.1 Hz), 7.18 (2H, d, *J* = 8.1 Hz), 7.00 (1H, s), 6.89 (1H, s), 6.32 (1H, s), 2.47 (3H, s), 2.39 (3H, s), 2.23 (3H, s), 2.06 (2H, t, *J* = 6.6 Hz), 1.28–1.41 (4H, m), 0.91 (3H, t, *J* = 7.1 Hz); ^13^C{^1^H} NMR (100 MHz, CDCl_3_) *δ*: 143.3, 137.5, 136.8, 136.7, 132.4, 132.1, 130.3, 129.1, 127.6, 121.8, 95.2, 76.4, 30.5, 22.0, 21.5, 20.7, 19.5, 19.1, 13.6; MS (ESI-TOF) *m*/*z*: [M + Na]^+^ 378; HRMS (ESI-TOF) *m*/*z*: [M + Na]^+^ Calcd for C_21_H_25_NNaO_2_S 378.1504; Found 378.1475.

*N*-(2,4-Dimethyl-6-(*p*-tolylethynyl)phenyl)-4-methoxybenzenesulfonamide (**1n**). In accordance with the experimental procedure for the synthesis of **1b**, **1n** was prepared from 2,4-dimethyl-6-(4-tolylethynyl)aniline (401 mg, 1.7 mmol) and 4-methoxybenzenesulfonyl chloride (386 mg, 1.9 mmol). The reaction was conducted for 20 h at 0 °C–rt. Purification of the residue by column chromatography (hexane/AcOEt = 20 and then 5) gave **1n** (536 mg, 78%). **1n**: white solid; mp 162–163 °C; IR (neat) 3239, 2207, 1335, 1155 cm^−1^; ^1^H NMR (400 MHz, CDCl_3_) *δ*: 7.53 (2H, dt, *J* = 9.0, 2.0 Hz), 7.18 (2H, d, *J* = 8.1 Hz), 7.12 (2H, d, *J* = 8.1 Hz), 7.06 (1H, s), 7.02 (1H, s), 6.66 (2H, dt, *J* = 9.0, 2.0 Hz), 6.41 (1H, s), 3.66 (3H, s), 2.50 (3H, s), 2.37 (3H, s), 2.27 (3H, s); ^13^C{^1^H} NMR (100 MHz, CDCl_3_) *δ*: 162.8, 138.7, 137.9, 136.9, 132.7, 132.4, 131.4, 131.1, 130.4, 129.6, 129.0, 121.5, 119.5, 113.8, 93.9, 84.9, 55.3, 21.5, 20.7, 19.4; MS (ESI-TOF) *m*/*z*: [M + Na]^+^ 428; HRMS (ESI-TOF) *m*/*z*: [M + Na]^+^ Calcd for C_24_H_23_NNaO_3_S 428.1296; Found 428.1270.

*N*-(2,4-Dimethyl-6-(*p*-tolylethynyl)phenyl)-4-nitrobenzenesulfonamide (**1o**). In accordance with the experimental procedure for the synthesis of **1a**, **1o** was prepared from 2,4-dimethyl-6-(4-tolylethynyl)aniline (489 mg, 2.1 mmol) and 4-nitrobenzenesulfonyl chloride (507 mg, 2.3 mmol). The reaction was conducted for 22 h at 0 °C–rt. Hexane was added to the residue and the mixture was filtered in vacuo. After washing the residue by hexane, **1o** was obtained (400 mg, 46%). **1o**: yellow solid; mp 203–204 °C; IR (neat) 3242, 2209, 1522, 1341, 1167 cm^−1^; ^1^H NMR (400 MHz, CDCl_3_) *δ*: 7.95 (2H, dt, *J* = 9.0, 2.0 Hz), 7.76 (2H, dt, *J* = 9.0, 2.0 Hz), 7.05–7.11 (3H, m), 7.03–7.05 (3H, m), 6.53 (1H, s), 2.54 (3H, s), 2.38 (3H, s), 2.30 (3H, s); ^13^C{^1^H} NMR (100 MHz, CDCl_3_) *δ*: 149.8, 145.2, 139.4, 138.5, 138.0, 133.0, 131.2, 131.1, 130.7, 129.2, 128.7, 123.8, 121.6, 118.8, 94.3, 84.7, 21.5, 20.8, 19.4; MS (ESI-TOF) *m*/*z*: [M + Na]^+^ 443; HRMS (ESI-TOF) *m*/*z*: [M + Na]^+^ Calcd for C_23_H_20_N_2_NaO_4_S 443.1042; Found 443.1015.

*N*-(2,4-Dimethyl-6-(phenylethynyl)phenyl)-4-nitrobenzenesulfonamide (**1p**). In accordance with the experimental procedure for the synthesis of **1b**, **1p** was prepared from 2,4-dimethyl-6-(phenylethynyl)aniline (426 mg, 1.9 mmol) and 4-nitrobenzenesulfonyl chloride (488 mg, 2.2 mmol). The reaction was conducted for 5 h at 0 °C–rt. Purification of the residue by column chromatography (hexane/AcOEt = 5) gave **1p** (575 mg, 73%). **1p**: white solid; mp 201–203 °C; IR (neat) 3231, 1522, 1344, 1167 cm^−1^; ^1^H NMR (400 MHz, CDCl_3_) *δ*: 7.95 (2H, d, *J* = 8.5 Hz), 7.77 (2H, d, *J* = 8.5 Hz), 7.29–7.38 (3H, m), 7.13–7.16 (3H, m), 7.06 (1H, s), 6.53 (1H, s), 2.54 (3H, s), 2.31 (3H, s); ^13^C{^1^H} NMR (100 MHz, CDCl_3_) *δ*: 149.8, 145.2, 138.6, 138.1, 133.2, 131.23, 131.18, 130.8, 129.1, 128.7, 128.5, 123.8, 121.9, 121.5, 94.0, 85.3, 20.8, 19.4; MS (ESI-TOF) *m*/*z*: [M + Na]^+^ 429; HRMS (ESI-TOF) *m*/*z*: [M + Na]^+^ Calcd for C_22_H_18_N_2_NaO_4_S 429.0885; Found 429.0869.

*N*-(2,4-dimethyl-6-(phenylethynyl)phenyl)benzenesulfonamide (**1q**). In accordance with the experimental procedure for the synthesis of **1b**, **1q** was prepared from 2,4-dimethyl-6-(phenylethynyl)aniline (289 mg, 1.3 mmol) and benzenesulfonyl chloride (255 mg, 1.4 mmol). The reaction was conducted for 18 h at 0 °C–rt. Purification of the residue by column chromatography (hexane/AcOEt = 15 and then 10) gave **1q** (417 mg, 88%). **1q**: white solid; mp 154–155 °C; IR (neat) 3248, 2211, 1327, 1159 cm^−1^; ^1^H NMR (400 MHz, CDCl_3_) *δ*: 7.62–7.64 (2H, m), 7.20–7.41 (8H, m), 7.08 (1H, s), 7.04 (1H, s), 6.48 (1H, s), 2.50 (3H, s), 2.28 (3H, s); ^13^C{^1^H} NMR (100 MHz, CDCl_3_) *δ*: 139.3, 137.9, 137.1, 132.9, 132.7, 132.2, 131.5, 130.4, 128.6, 128.5, 128.2, 127.5, 122.4, 121.4, 93.9, 85.3, 20.7, 19.4; MS (ESI-TOF) *m*/*z*: [M + Na]^+^ 384; HRMS (ESI-TOF) *m*/*z*: [M + Na]^+^ Calcd for C_22_H_19_NNaO_2_S 384.1034; Found 384.1011.

*N*-(2,4-Dimethyl-6-(p-tolylethynyl)phenyl)methanesulfonamide (**1r**). In accordance with the experimental procedure for the synthesis of **1b**, **1r** was prepared from 2,4-dimethyl-6-(4-tolylethynyl)aniline (457 mg, 1.9 mmol) and methanesulfonyl chloride (253 mg, 2.2 mmol). The reaction was conducted for 23 h at 0 °C–rt. Purification of the residue by column chromatography (hexane/AcOEt = 20 and then 5) gave **1r** (430 mg, 71%). **1r**: white solid; mp 171–177 °C; IR (neat) 3246, 2199, 1316, 1157 cm^−1^; ^1^H NMR (400 MHz, CDCl_3_) *δ*: 7.42 (2H, d, *J* = 8.1 Hz), 7.24 (1H, s), 7.19 (2H, d, *J* = 8.1 Hz), 7.09 (1H, s), 6.35 (1H, s), 3.08 (3H, s), 2.47 (3H, s), 2.39 (3H, s), 2.32 (3H, s); ^13^C{^1^H} NMR (100 MHz, CDCl_3_) *δ*: 139.3, 138.5, 137.6, 133.0, 132.2, 131.3, 130.7, 129.4, 121.4, 119.0, 95.0, 85.7, 40.5, 21.5, 20.7, 19.3; MS (ESI-TOF) *m*/*z*: [M + Na]^+^ 336; HRMS (ESI-TOF) *m*/*z*: [M + Na]^+^ Calcd for C_18_H_19_NNaO_2_S 336.1034; Found 336.1017.

*N*-(2,4-Dimethyl-6-(p-tolylethynyl)phenyl)-2,4,6-trimethylbenzenesulfonamide (**1s**). In accordance with the experimental procedure for the synthesis of **1b**, **1r** was prepared from 2,4-dimethyl-6-(4-tolylethynyl)aniline (339 mg, 1.4 mmol) and 2,4,6-trimethylbenzenesulfonyl chloride (313 mg, 1.4 mmol). The reaction was conducted for 16 h at 0 °C–rt. Purification of the residue by column chromatography (hexane/AcOEt = 30 and then 20) gave **1s** (443 mg, 74%). **1s**: white solid; mp 161–162 °C; IR (neat) 3277, 2207, 1323, 1161 cm^−1^; ^1^H NMR (400 MHz, CDCl_3_) *δ*: 7.18 (2H, d, *J* = 8.1 Hz), 7.12 (2H, d, *J* = 8.1 Hz), 7.09 (1H, s), 7.03 (1H, s), 6.70 (2H, s), 6.43 (1H, s), 2.38 (3H, s), 2.36 (6H, s), 2.34 (3H, s), 2.27 (3H, s), 2.17 (3H, s); ^13^C{^1^H} NMR (100 MHz, CDCl_3_) *δ*: 141.9, 139.2, 138.6, 137.9, 137.0, 135.2, 132.6, 132.4, 131.9, 131.4, 130.7, 128.8, 122.1, 119.6, 93.6, 84.8, 23.5, 21.5, 20.8, 20.7, 19.2; MS (ESI-TOF) *m*/*z*: [M + Na]^+^ 440; HRMS (ESI-TOF) *m*/*z*: [M + Na]^+^ Calcd for C_26_H_27_NNaO_2_S 440.1660; Found 440.1674.

*N*-Allyl-*N*-[(2-bromo-4-methyl-6-phenyl)phenyl]-4-methylbenzenesulfonamide (**2c**). Under N_2_ atmosphere, to **1c** (125 mg, 0.3 mmol) in THF (2.5 mL) was added NaH (60% assay, 12 mg, 0.3 mmol) at 0 °C, and the mixture was stirred for 20 min at −20 °C. (Allyl-Pd-Cl)_2_ (2.5 mg, 0.0068 mmol), (*S*,*S*)-Trost ligand (10.5 mg, 0.0133 mmol) and allyl acetate (98 μL, 0.9 mmol) in THF (2.0 mL) were added to the reaction mixture, and then the mixture was stirred for 10 h at −20 °C. The mixture was poured into 1N HCl solution and extracted with AcOEt. The AcOEt extracts were washed with brine, dried over MgSO_4_, and evaporated to dryness. Purification of the residue by column chromatography (hexane/AcOEt = 10) gave **2c** (133 mg, 97%). The ee (38% ee) of **2c** was determined by HPLC analysis using a chiral column (CHIRALCEL OD-3) (25 cm × 0.46 cm i.d.; 15% *i*-PrOH in hexane; flow rate, 0.8 mL/min; (+)-**2c** (major); *t*_R_ = 8.0 min, (-)-**2c** (minor); *t*_R_ = 9.5 min). **2c**: white solid; mp 113–115 °C (38% ee); IR (neat) 1333, 1150 cm^−1^; [α]_D_ = +18.6° (38% ee, CHCl_3_, c = 1.00); ^1^H NMR (400 MHz, CDCl_3_) *δ*: 7.52 (2H, dt, *J* = 8.6, 1.9 Hz), 7.32–7.45 (6H, m), 7.17 (2H, d, *J* = 8.1 Hz), 7.07 (1H, d, *J* = 2.9 Hz), 5.68 (1H, ddt, *J* = 17.1, 10.0, 7.1 Hz), 5.08 (1H, dd, *J* = 17.1, 1.4 Hz), 5.01 (1H, dd, *J* = 10.2, 1.4 Hz), 4.19 (1H, dd, *J* = 14.2, 7.1 Hz), 3.96 (1H, dd, *J* = 14.2, 7.1 Hz), 2.40 (3H, s), 2.35 (3H, s); ^13^C{^1^H} NMR (100 MHz, CDCl_3_) *δ*: 146.2, 142.9, 139.6, 139.4, 137.6, 133.7, 133.6, 132.5, 131.7, 129.5, 129.0, 128.0, 127.6, 127.4, 125.9, 119.1, 53.6, 21.4, 20.6; MS (ESI-TOF) *m*/*z*: [M + Na]^+^ 480; HRMS (ESI-TOF) *m*/*z*: [M + Na]^+^ Calcd for C_23_H_22_^81^BrNNaO_2_S 480.0432; Found 480.0417.

*N*-Allyl-*N*-(2-(*tert*-butyl)-6-methylphenyl)-4-methylbenzenesulfonamide (**2a**). In accordance with the experimental procedure for the synthesis of **2c**, **2a** was prepared from **1a** (96 mg, 0.3 mmol). The reaction was conducted for 21 h at −20 °C. Purification of the residue by column chromatography (hexane/AcOEt = 30) gave **2a** (62 mg, 58%). The ee (10% ee) of **2a** was determined by HPLC analysis using a chiral column (CHIRALPAK AS-H) (25 cm × 0.46 cm i.d.; 15% *i*-PrOH in hexane; flow rate, 0.8 mL/min; (-)-**2a** (major); *t*_R_ = 12.8 min, (+)-**2a** (minor); *t*_R_ = 8.7 min). **2a**: white solid; mp 105–107 °C (10% ee); IR (neat) 1339, 1159 cm^−1^; [α]_D_ = −5.2° (10% ee, CHCl_3_, c = 0.83); ^1^H NMR (400 MHz, CDCl_3_) *δ*: 7.63 (2H, d, *J* = 7.6 Hz), 7.44–7.46 (1H, m), 7.29 (2H, d, *J* = 7.6 Hz), 7.17 (1H, t, *J* = 7.6 Hz), 6.87 (1H, dd, *J* = 7.6, 0.9 Hz), 5.66 (1H, dddd, *J* = 16.6, 10.4, 7.6, 6.2 Hz), 5.21 (1H, dd, *J* = 16.6, 1.4 Hz), 5.04 (1H, dd, *J* = 10.4, 1.4 Hz), 4.50 (1H, ddt, *J* = 13.2, 6.2, 1.4 Hz), 4.14 (1H, dd, *J* = 13.2, 7.6 Hz), 2.43 (3H, s), 1.55 (9H, s), 1.47 (3H, s); ^13^C{^1^H} NMR (100 MHz, CDCl_3_) *δ*: 151.7, 143.2, 138.9, 137.3, 133.7, 131.5, 129.5, 128.7, 128.6, 127.9, 127.7, 119.3, 53.8, 37.2, 33.3, 21.5, 19.4; MS (ESI-TOF) *m*/*z*: [M + Na]^+^ 380; HRMS (ESI-TOF) *m*/*z*: [M + Na]^+^ Calcd for C_21_H_27_NNaO_2_S 380.1660; Found 380.1641.

*N*-Allyl-*N*-((2-iodo-4-methyl-6-phenyl)phenyl)-4-methylbenzenesulfonamide (**2b**). In accordance with the experimental procedure for the synthesis of **2c**, **2b** was prepared from **1b** (93 mg, 0.2 mmol). The reaction was conducted for 27 h at −20 °C. Purification of the residue by column chromatography (hexane/AcOEt = 10) gave **2b** (10 mg, 10%). The ee (43% ee) of **2b** was determined by HPLC analysis using a chiral column (CHIRALCEL OD-3) (25 cm × 0.46 cm i.d.; 15% *i*-PrOH in hexane; flow rate, 0.8 mL/min; (+)-**2b** (major); *t*_R_ = 9.0 min, (-)-**2b** (minor); *t*_R_ = 10.0 min). **2b**: colorless oil; IR (neat) 1344, 1157 cm^−1^; [α]_D_ = +32.8° (48% ee, CHCl_3_, c = 1.01); ^1^H NMR (400 MHz, CDCl_3_) *δ*: 7.76 (1H, d, *J* = 1.4 Hz), 7.53 (2H, d, 8.1 Hz), 7.28–7.36 (5H, m), 7.16 (2H, d, *J* = 8.1 Hz), 7.07 (1H, d, *J* = 1.4 Hz), 5.77 (1H, ddt, *J* = 17.1, 10.0, 7.1 Hz), 5.14 (1H, dd, *J* = 17.1, 1.4 Hz), 5.04 (1H, dd, *J* = 10.0, 1.4 Hz), 4.26 (1H, dd, *J* = 14.5, 6.9 Hz), 3.99 (1H, dd, *J* = 14.5, 7.1 Hz), 2.40 (3H, s), 2.32 (3H, s); ^13^C{^1^H} NMR (100 MHz, CDCl_3_) *δ*: 145.7, 142.9, 140.5, 139.7, 139.6, 137.9, 137.3, 132.7, 129.4, 129.0, 128.1, 127.43, 127.38, 119.1, 102.5, 54.0, 21.4, 20.3; MS (ESI-TOF) *m*/*z*: [M +Na]^+^ 526; HRMS (ESI-TOF) *m*/*z*: [M + Na]^+^ Calcd for C_23_H_22_^127^INNaO_2_S 526.0314; Found 526.0294.

*N*-Allyl-*N*-(2-iodo-4,6-dimethylphenyl)-4-methylbenzenesulfonamide (**2d**). In accordance with the experimental procedure for the synthesis of **2c**, **2d** was prepared from **1d** (120 mg, 0.3 mmol). The reaction was conducted for 6 h at −20 °C. Purification of the residue by column chromatography (hexane/AcOEt = 5) gave **2d** (124 mg, 94%). The ee (65% ee) of **2d** was determined by HPLC analysis using a chiral column (CHIRALCEL OD-3) (25 cm × 0.46 cm i.d.; 3% *i*-PrOH in hexane; flow rate, 0.8 mL/min; (+)-**2d** (major); *t*_R_ = 11.1min, (-)-**2d** (minor); *t*_R_ = 10.5 min). **2d**: white solid; mp 67–69 °C (65% ee); IR (neat) 1343, 1159 cm^−1^; [α]_D_ = +40.9° (65% ee, CHCl_3_, c = 1.00); ^1^H NMR (400 MHz, CDCl_3_) *δ*: 7.74 (2H, d, *J* = 8.1 Hz), 7.50 (1H, d, *J* = 1.4 Hz), 7.29 (2H, d, *J* = 8.1 Hz), 7.01 (1H, d, *J* = 1.4 Hz), 5.97 (1H, m), 5.02–5.08 (2H, m), 4.32 (1H, dd, *J* = 14.5, 6.6 Hz), 4.12 (1H, dd, *J* = 14.5, 7.8 Hz), 2.42 (3H, s), 2.28 (3H, s), 2.23 (3H, s); ^13^C{^1^H} NMR (100 MHz, CDCl_3_) *δ*: 143.3, 141.8, 139.8, 138.7, 138.3, 137.5, 132.8, 132.1, 129.4, 128.0, 119.3, 101.1, 53.6, 21.5, 20.5, 20.3; MS (ESI-TOF) *m*/*z*: [M + Na]^+^ 464; HRMS (ESI-TOF) *m*/*z*: [M + Na]^+^ Calcd for C_18_H_20_^127^INNaO_2_S 464.0157; Found 464.0128.

(*E*)-*N*-Allyl-*N*-(2,4-dimethyl-6-styrylphenyl)-4-methylbenzenesulfonamide (**2e**). In accordance with the experimental procedure for the synthesis of **2c**, **2e** was prepared from **1e** (113 mg, 0.3 mmol). The reaction was conducted for 23 h at −20 °C. Purification of the residue by column chromatography (hexane/AcOEt = 5) gave **2e** (114 mg, 91%). The ee (65% ee) of **2e** was determined by HPLC analysis using a chiral column (CHIRALCEL OD-3) (25 cm × 0.46 cm i.d.; 15% *i*-PrOH in hexane; flow rate, 0.8 mL/min; (+)-**2e** (major); *t*_R_ = 7.6 min, (-)-**2e** (minor); *t*_R_ = 6.7 min). **2e**: white oil; IR (neat) 1343, 1157 cm^−1^; [α]_D_ = +132.8° (65% ee, CHCl_3_, c = 1.01); ^1^H NMR (400 MHz, CDCl_3_) *δ*: 7.74 (2H, dd, *J* = 8.5, 1.9 Hz), 7.32 (1H, s), 7.20–7.27 (5H, m), 6.99–7.04 (3H, m), 6.86 (1H, d, *J* = 16.1 Hz), 6.50 (1H, d, *J* = 16.1 Hz), 5.90 (1H, dddd, *J* = 16.6, 10.0, 7.6, 6.2 Hz), 4.99–5.04 (2H, m), 4.32 (1H, ddt, *J* = 14.8, 6.2, 1.0 Hz), 3.97 (1H, dd, *J* = 14.8, 7.6 Hz), 2.39 (3H, s), 2.35 (3H, s), 2.30 (3H, s); ^13^C{^1^H} NMR (100 MHz, CDCl_3_) *δ*: 143.4, 140.5, 138.2, 138.1, 137.0, 136.5, 133.8, 132.8, 131.6, 129.9, 129.7, 128.4, 127.6, 127.5, 126.5, 125.2, 124.2, 119.2, 54.4, 21.4, 21.2, 19.6; MS (ESI-TOF) *m*/*z*: [M + Na]^+^ 440; HRMS (ESI-TOF) *m*/*z*: [M + Na]^+^ Calcd for C_26_H_27_NNaO_2_S 440.1660; Found 440.1648.

*N*-Allyl-*N*-(2-iodo-4-methyl-6-(*p*-tolylethynyl)phenyl)-4-methylbenzenesulfonamide (**2f**). In accordance with the experimental procedure for the synthesis of **2c**, **2f** was prepared from **1f** (94 mg, 0.19 mmol). The reaction was conducted for 5 h at −20 °C. Purification of the residue by column chromatography (hexane/AcOEt = 10) gave **2f** (84 mg, 82%). The ee (73% ee) of **2f** was determined by HPLC analysis using a chiral column (CHIRALCEL OD-3) (25 cm × 0.46 cm i.d.; 3% *i*-PrOH in hexane; flow rate, 0.8 mL/min; (+)-**2f** (major); *t*_R_ = 24.1 min, (-)-**2f** (minor); *t*_R_ = 27.7 min). **2f**: colorless oil; IR (neat) 2205, 1344, 1157 cm^−1^; [α]_D_ = +71.1° (74% ee, CHCl_3_, c = 0.83); ^1^H NMR (400 MHz, CDCl_3_) *δ*: 7.74 (2H, d, *J* = 8.1 Hz), 7.72 (1H, m), 7.28 (1H, d, *J* = 1.4 Hz),7.04–7.08 (6H, m), 6.12 (1H, dddd, *J* = 16.1, 10.0, 8.1, 6.2 Hz), 5.10 (1H, d, *J* = 17.1 Hz), 5.03 (1H, d, *J* = 10.0 Hz), 4.48 (1H, dd, *J* = 14.2, 6.2 Hz), 4.38 (1H, dd, *J* = 14.2, 8.1 Hz), 2.35 (3H, s), 2.28 (3H, s), 2.10 (3H, s); ^13^C{^1^H} NMR (100 MHz, CDCl_3_) *δ*: 143.2, 140.8, 139.63, 139.58, 138.7, 138.0, 134.2, 132.8, 131.3, 129.3, 128.7, 128.0, 125.2, 119.3, 119.2, 105.2, 94.5, 85.9, 53.1, 21.5, 21.2, 20.2; MS (ESI-TOF) *m*/*z*: [M + Na]^+^ 564; HRMS (ESI-TOF) *m*/*z*: [M + Na]^+^ Calcd for C_26_H_24_INNaO_2_S 564.0470; Found 564.0442.

*N*-Allyl-*N*-(2-bromo-4-methyl-6-(*p*-tolylethynyl)phenyl)-4-methylbenzenesulfonamide (**2g**). In accordance with the experimental procedure for the synthesis of **2c**, **2g** was prepared from **1g** (91 mg, 0.2 mmol). The reaction was conducted for 7 h at −20 °C. Purification of the residue by column chromatography (hexane/AcOEt = 5) gave **2g** (101 mg, quant). The ee (72% ee) of **2g** was determined by HPLC analysis using a chiral column (CHIRALCEL OD-3) (25 cm × 0.46 cm i.d.; 5% *i*-PrOH in hexane; flow rate, 0.8 mL/min; (+)-**2g** (major); *t*_R_ = 16.0 min, (-)-**2g** (minor); *t*_R_ = 13.5 min). **2g**: white oil; IR (neat) 2212, 1343, 1155 cm^−1^; [α]_D_ = +61.5° (69% ee, CHCl_3_, c = 0.68); ^1^H NMR (400 MHz, CDCl_3_) *δ*: 7.77 (2H, d, *J* = 8.1 Hz), 7.42 (1H, d, *J* = 1.9 Hz), 7.27 (1H, d, *J* = 1.9 Hz), 7.07–7.12 (6H, m), 6.06 (1H, dddd, *J* = 17.1, 10.0, 7.6, 6.6 Hz), 5.08 (1H, dd, *J* = 17.1, 1.0 Hz), 5.02 (1H, dd, *J* = 10.0, 1.0 Hz), 4.40 (1H, dd, *J* = 14.2, 6.6 Hz), 4.36 (1H, dd, *J* = 14.2, 7.6 Hz), 2.36 (3H, s), 2.31 (3H, s), 2.15 (3H, s); ^13^C{^1^H} NMR (100 MHz, CDCl_3_) *δ*: 143.1, 139.5, 138.7, 138.0, 136.4, 134.1, 133.0, 132.8, 131.3, 129.2, 128.7, 127.8, 127.4, 126.5, 119.3, 119.0, 94.6, 85.9, 52.9, 21.4, 21.2, 20.5; MS (ESI-TOF) *m*/*z*: [M + Na]^+^ 518; HRMS (ESI-TOF) *m*/*z*: [M + Na]^+^ Calcd for C_26_H_24_^81^BrNNaO_2_S 518.0588; Found 518.0592.

*N*-Allyl-*N*-(2-chloro-6-(*p*-tolylethynyl)phenyl)-4-methylbenzenesulfonamide (**2h**). In accordance with the experimental procedure for the synthesis of **2c**, **2h** was prepared from **1h** (119 mg, 0.3 mmol). The reaction was conducted for 5 h at −20 °C. Purification of the residue by column chromatography (hexane/AcOEt = 10) gave **2h** (124 mg, 95%). The ee (79% ee) of **2h** was determined by HPLC analysis using a chiral column (CHIRALPAK AS-H) (25 cm × 0.46 cm i.d.; 15% *i*-PrOH in hexane; flow rate, 0.8 mL/min; (+)-**2h** (major); *t*_R_ = 24.7 min, (-)-**2h** (minor); *t*_R_ = 18.6 min). **2h**: yellow solid; mp 101–104 °C (76% ee); IR (neat) 2224, 1346, 1155 cm^−1^; [α]_D_ = +103.8° (76% ee, CHCl_3_, c = 1.00); ^1^H NMR (400 MHz, CDCl_3_) *δ*: 7.78 (2H, d, *J* = 8.1 Hz), 7.40–7.44 (2H, m), 7.23 (1H, t, *J* = 7.6 Hz), 7.09–7.16 (6H, m), 6.01 (1H, ddt, *J* = 17.1, 10.0, 7.1 Hz), 5.05 (1H, d, *J* = 17.1 Hz), 5.01 (2H, d, *J* = 6.6 Hz), 4.01 (1H, d, *J* = 10.0 Hz), 2.36 (3H, s), 2.19 (3H, s); ^13^C{^1^H} NMR (100 MHz, CDCl_3_) *δ*: 143.2, 138.9, 138.0, 137.6, 137.2, 132.8, 131.7, 131.5, 130.2, 129.3, 129.0, 128.8, 127.9, 127.4, 119.3, 119.1, 95.1, 85.8, 52.8, 21.5, 21.3; MS (ESI-TOF) *m*/*z*: [M + Na]^+^ 458; HRMS (ESI-TOF) *m*/*z*: [M + Na]^+^ Calcd for C_25_H_22_^35^ClNNaO_2_S 458.0958; Found 458.0951.

*N*-Allyl-*N*-(2,4-dimethyl-6-(*p*-tolylethynyl)phenyl)-4-methylbenzenesulfonamide (**2i**). In accordance with the experimental procedure for the synthesis of **2c**, **2i** was prepared from **1i** (117 mg, 0.3 mmol). The reaction was conducted for 6 h at −20 °C. Purification of the residue by column chromatography (hexane/AcOEt = 20) gave **2i** (109 mg, 88%). The ee (86% ee) of **2i** was determined by HPLC analysis using a chiral column (CHIRALPAK AD-H) (25 cm × 0.46 cm i.d.; 15% *i*-PrOH in hexane; flow rate, 0.8 mL/min; (+)-**2i** (major); *t*_R_ = 9.9 min, (-)-**2i** (minor); *t*_R_ = 12.1 min). **2i**: yellow oil; IR (neat) 2205, 1341, 1155 cm^−1^; [α]_D_ = +180.7° (80% ee, CHCl_3_, c = 1.01); ^1^H NMR (400 MHz, CDCl_3_) *δ*: 7.71 (2H, dt, *J* = 8.5, 1.9 Hz), 7.14 (1H, d, *J* = 1.9 Hz), 7.02–7.07 (5H, m), 6.95 (2H, d, *J* = 8.1 Hz), 5.98 (1H, dddd, *J* = 17.2, 10.4, 8.5, 5.7 Hz), 5.07 (1H, dd, *J* = 17.2, 1.4 Hz), 5.04 (1H, d, *J* = 10.4 Hz), 4.48 (1H, ddt, *J* = 14.2, 5.7, 1.4 Hz), 4.25 (1H, dd, *J* = 14.2, 8.5 Hz), 2.45 (3H, s), 2.35 (3H, s), 2.30 (3H, s), 2.08 (3H, s); ^13^C{^1^H} NMR (100 MHz, CDCl_3_) *δ*: 143.0, 141.2, 138.3, 137.83, 137.78, 136.2, 133.2, 132.2, 131.8, 131.1, 129.3, 128.6, 127.8, 123.3, 119.7, 119.0, 93.3, 86.8, 53.0, 21.4, 21.2, 20.8, 19.6; MS (ESI-TOF) *m*/*z*: [M + Na]^+^ 452; HRMS (ESI-TOF) *m*/*z*: [M + Na]^+^ Calcd for C_27_H_27_NNaO_2_S 452.1660; Found 452.1631.

*N*-Allyl-*N*-(2-((4-methoxyphenyl)ethynyl)-4,6-dimethylphenyl)-4-methylbenzenesulfonamide (**2j**). In accordance with the experimental procedure for the synthesis of **2c**, **2j** was prepared from **1j** (122 mg, 0.3 mmol). The reaction was conducted for 23 h at −20 °C. Purification of the residue by column chromatography (hexane/AcOEt = 20) gave **2j** (131 mg, 98%). The ee (88% ee) of **2j** was determined by HPLC analysis using a chiral column (CHIRALPAK AD-H) (25 cm × 0.46 cm i.d.; 15% *i*-PrOH in hexane; flow rate, 0.8 mL/min; (+)-**2j** (major); *t*_R_ = 17.7 min, (-)-**2j** (minor); *t*_R_ = 23.6 min). **2j**: white solid; mp 78–80 °C (87% ee); IR (neat) 2211, 1341, 1159 cm^−1^; [α]_D_ = +193.3° (87% ee, CHCl_3_, c = 1.00); ^1^H NMR (400 MHz, CDCl_3_) *δ*: 7.72 (2H, d, *J* = 8.1 Hz), 7.13 (1H, s), 7.06 (1H, s), 7.05 (2H, d, *J* = 8.1 Hz), 7.00 (2H, dt, *J* = 8.5, 1.9 Hz), 6.77 (2H, dt, *J* = 8.5, 2.8 Hz), 5.98 (1H, dddd, *J* = 18.0, 10.4, 8.5, 5.7 Hz), 5.07 (1H, d, *J* = 18.0 Hz), 5.04 (1H, d, *J* = 10.4 Hz), 4.47 (1H, dd, *J* = 14.2, 5.7 Hz), 4.25 (1H, dd, *J* = 14.2, 8.5 Hz), 3.81 (3H, s), 2.44 (3H, s), 2.29 (3H, s), 2.11 (3H, s); ^13^C{^1^H} NMR (100 MHz, CDCl_3_) *δ*: 159.4, 142.9, 141.1, 137.9, 137.7, 136.0, 133.1, 132.6, 131.9, 131.6, 129.2, 127.7, 123.4, 118.9, 114.9, 113.5, 93.2, 86.1, 55.2, 52.9, 21.2, 20.7, 19.5; MS (ESI-TOF) *m*/*z*: [M + Na]^+^ 468; HRMS (ESI-TOF) *m*/*z*: [M + Na]^+^ Calcd for C_27_H_27_NNaO_3_S 468.1609; Found 468.1615.

*N*-Allyl-*N*-(2,4-dimethyl-6-(phenylethynyl)phenyl)-4-methylbenzenesulfonamide (**2k**). In accordance with the experimental procedure for the synthesis of **2c**, **2j** was prepared from **1j** (113 mg, 0.3 mmol). The reaction was conducted for 21 h at −20 °C. Purification of the residue by column chromatography (hexane/AcOEt = 20) gave **2j** (114 mg, 92%). The ee (89% ee) of **2j** was determined by HPLC analysis using a chiral column (CHIRALPAK AD-H) (25 cm × 0.46 cm i.d.; 15% *i*-PrOH in hexane; flow rate, 0.8 mL/min; (+)-**2k** (major); *t*_R_ = 11.8 min, (-)-**2k** (minor); *t*_R_ = 16.4 min). **2k**: white solid; mp 87–89 °C (90% ee); IR (neat) 1339, 1155 cm^−1^; [α]_D_ = +196.7° (90% ee, CHCl_3_, c = 0.79); ^1^H NMR (400 MHz, CDCl_3_) *δ*: 7.73 (2H, dt, *J* = 8.5, 1.9 Hz), 7.23–7.29 (3H, m), 7.16 (1H, d, *J* = 2.4 Hz), 7.06–7.09 (3H, m), 7.03 (2H, d, *J* = 8.1 Hz), 6.00 (1H, dddd, *J* = 17.1, 10.0, 8.5, 5.7 Hz), 5.09 (1H, d, *J* = 17.1 Hz), 5.07 (1H, d, *J* = 10.0 Hz), 4.51 (1H, ddt, *J* = 14.2, 5.7, 1.4 Hz), 4.28 (1H, dd, *J* = 14.2, 8.5 Hz), 2.46 (3H, s), 2.31 (3H, s), 2.07 (3H, s); ^13^C{^1^H} NMR (100 MHz, CDCl_3_) *δ*: 143.0, 141.2, 137.8, 136.2, 133.1, 132.3, 131.9, 131.1, 129.3, 128.1, 127.8, 127.7, 123.1, 122.7, 119.0, 93.0, 87.3, 53.0, 21.1, 20.7, 19.5; MS (ESI-TOF) *m*/z: [M + Na]^+^ 438; HRMS (ESI-TOF) *m*/*z*: [M + Na]^+^ Calcd for C_26_H_25_NNaO_2_S 438.1504; Found 438.1484.

*N*-Allyl-*N*-(2,4-dimethyl-6-((trimethylsilyl)ethynyl)phenyl)-4-methylbenzenesulfonamide (**2l**). In accordance with the experimental procedure for the synthesis of **2c**, **2l** was prepared from **1l** (112 mg, 0.3 mmol). The reaction was conducted for 9 h at −20 °C. Purification of the residue by column chromatography (hexane/AcOEt = 5) gave **2l** (86 mg, 70%). The ee (75% ee) of **2l** was determined by HPLC analysis using a chiral column (CHIRALPAK AD-H) (25 cm × 0.46 cm i.d.; 15% *i*-PrOH in hexane; flow rate, 0.8 mL/min; (+)-**2l** (major); *t*_R_ = 5.3 min, (-)-**2l** (minor); *t*_R_ = 6.1 min). **2l**: yellow oil; IR (neat) 2153, 1344, 1159 cm^−1^; [α]_D_ = +129.7° (75% ee, CHCl_3_, c = 0.83); ^1^H NMR (400 MHz, CDCl_3_) *δ*: 7.68 (2H, dt, *J* = 8.5, 1.9 Hz), 7.25 (2H, d, *J* = 8.5 Hz), 7.10 (1H, d, *J* = 1.4 Hz), 7.05 (1H, m), 5.93 (1H, dddd, *J* = 17.1, 10.0, 8.1, 5.7 Hz), 5.04 (1H, dt, *J* = 17.1, 1.4 Hz), 5.01 (1H, d, *J* = 10.0 Hz), 4.34 (1H, ddt, *J* = 14.7, 5.7, 1.4 Hz), 4.18 (1H, dd, *J* = 14.7, 8.1 Hz), 2.41 (3H, s), 2.36 (3H, s), 2.26 (3H, s), 0.02–0.03 (9H, m); ^13^C{^1^H} NMR (100 MHz, CDCl_3_) *δ*: 142.8, 141.0, 138.1, 137.7, 136.3, 133.3, 132.6, 129.4, 128.1, 123.3, 118.8, 102.7, 98.4, 52.8, 21.6, 20.7, 19.6, −0.38; MS (ESI-TOF) *m*/*z*: [M + Na]^+^ 434; HRMS (ESI-TOF) *m*/*z*: [M + Na]^+^ Calcd for C_23_H_29_NNaO_2_S^28^Si 434.1586; Found 434.1560.

*N*-Allyl-*N*-(2-(hex-1-yn-1-yl)-4,6-dimethylphenyl)-4-methylbenzenesulfonamide (**2m**). In accordance with the experimental procedure for the synthesis of **2c**, **2m** was prepared from **1m** (107 mg, 0.3 mmol). The reaction was conducted for 25 h at −20 °C. Purification of the residue by column chromatography (hexane/AcOEt = 20) gave **2m** (114 mg, 96%). The ee (77% ee) of **2m** was determined by HPLC analysis using a chiral column (CHIRALPAK AS-H) (25 cm × 0.46 cm i.d.; 15% *i*-PrOH in hexane; flow rate, 0.8 mL/min; (+)-**2m** (major); *t*_R_ = 8.5 min, (-)-**2m** (minor); *t*_R_ = 6.6 min). **2m**: yellow oil; IR (neat) 2230, 1344, 1159 cm^−1^; [α]_D_ = +110.6° (77% ee, CHCl_3_, c = 0.87); ^1^H NMR (400 MHz, CDCl_3_) *δ*: 7.71 (2H, dt, *J* = 8.5, 1.9 Hz), 7.26 (2H, d, *J* = 8.5 Hz), 7.00 (1H, s), 6.99 (1H, s), 5.91 (1H, dddd, *J* = 17.1, 10.0, 8.5, 5.7 Hz), 4.99–5.05 (2H, m), 4.40 (1H, ddt, *J* =14.2, 5.7, 1.0 Hz), 4.12 (1H, dd, *J* = 14.2, 8.5 Hz), 2.41 (3H, s), 2.40 (3H, s), 2.25 (3H, s), 1.80–1.92 (2H, m), 1.19–1.34 (4H, m), 0.87 (3H, t, *J* = 7.1 Hz); ^13^C{^1^H} NMR (100 MHz, CDCl_3_) *δ*: 142.6, 141.0, 138.3, 137.6, 136.1, 133.3, 131.9, 131.5, 129.1, 128.0, 123.7, 118.7, 94.4, 78.2, 52.9, 30.4, 22.0, 21.4, 20.7, 19.6, 18.9, 13.5; MS (ESI-TOF) *m*/*z*: [M + Na]^+^ 418; HRMS (ESI-TOF) *m*/*z*: [M + Na]^+^ Calcd for C_24_H_29_NNaO_2_S 418.1817; Found 418.1843.

*N*-Allyl-*N*-(2,4-dimethyl-6-(*p*-tolylethynyl)phenyl)-4-methoxybenzenesulfonamide (**2n**). In accordance with the experimental procedure for the synthesis of **2c**, **2n** was prepared from **1n** (122 mg, 0.3 mmol). The reaction was conducted for 22 h at −20 °C. Purification of the residue by column chromatography (hexane/AcOEt = 20 and then 10) gave **2n** (117 mg, 88%). The ee (85% ee) of **2n** was determined by HPLC analysis using a chiral column (CHIRALPAK AS-H) (25 cm × 0.46 cm i.d.; 5% *i*-PrOH in hexane; flow rate, 0.8 mL/min; (+)-**2n** (major); *t*_R_ = 29.5 min, (-)-**2j** (minor); *t*_R_ = 24.9 min). **2n**: white solid; mp 124–132 °C (84% ee); IR (neat) 2207, 1344, 1150 cm^−1^; [*α*]_D_ = +194.8° (84% ee, CHCl_3_, c = 0.99); ^1^H NMR (400 MHz, CDCl_3_) *δ*: 7.75 (2H, d, *J* = 8.5 Hz), 7.13 (1H, s), 7.07 (1H, s), 7.05 (2H, d, *J* = 8.1 Hz), 6.97 (2H, d, *J* = 8.1 Hz), 6.68 (2H, d, *J* = 8.5 Hz), 5.97 (1H, dddd, *J* = 17.1, 10.0, 8.5, 5.7 Hz), 5.07 (1H, d, *J* = 18.5 Hz), 5.04 (1H, d, *J* = 10.4 Hz), 4.46 (1H, dd, *J* = 14.5, 5.7 Hz), 4.24 (1H, dd, *J* = 14.5, 8.1 Hz), 3.52 (3H, s), 2.44 (3H, s), 2.35 (3H, s), 2.30 (3H, s); ^13^C{^1^H} NMR (100 MHz, CDCl_3_) *δ*: 162.4, 141.3, 138.3, 137.8, 136.3, 133.3, 132.5, 132.2, 131.8, 131.1, 129.9, 128.7, 123.3, 119.8, 119.0, 113.8, 93.2, 86.9, 55.0, 53.0, 21.5, 20.8, 19.6; MS (ESI-TOF) *m*/*z*: [M + Na]^+^ 468; HRMS (ESI-TOF) *m*/*z*: [M + Na]^+^ Calcd for C_27_H_27_NNaO_3_S 468.1609; Found 468.1581.

*N*-Allyl-*N*-(2,4-dimethyl-6-(*p*-tolylethynyl)phenyl)-4-nitrobenzenesulfonamide (**2o**). In accordance with the experimental procedure for the synthesis of **2c**, **2o** was prepared from **1o** (126 mg, 0.3 mmol). The reaction was conducted for 6 h at −20 °C. Purification of the residue by column chromatography (hexane/AcOEt = 10) gave **2o** (137 mg, quant). The ee (89% ee) of **2o** was determined by HPLC analysis using a chiral column (CHIRALCEL OD-3) (25 cm × 0.46 cm i.d.; 15% *i*-PrOH in hexane; flow rate, 0.8 mL/min; (+)-**2o** (major); *t*_R_ = 9.9 min, (-)-**2o** (minor); *t*_R_ = 8.0 min). **2o**: yellow solid; mp 102–104 °C (97% ee); IR (neat) 2209, 1524, 1344, 1159 cm^−1^; [α]_D_ = +201.0° (97% ee, CHCl_3_, c = 0.59); ^1^H NMR (400 MHz, CDCl_3_) *δ*: 7.96 (2H, dd, *J* = 6.9, 2.4 Hz), 7.91 (2H, dd, *J* = 6.9, 2.4 Hz), 7.11 (1H, s), 7.10 (1H, s), 6.99 (2H, d, *J* = 8.1 Hz), 6.86 (2H, d, *J* = 8.1 Hz), 5.98 (1H, dddd, *J* = 17.1, 10.0, 8.5, 5.7 Hz), 5.13 (1H, d, *J* = 17.1 Hz), 5.11 (1H, d, *J* = 10.0 Hz), 4.54 (1H, dd, *J* = 14.2, 5.7 Hz), 4.27 (1H, dd, *J* = 14.2, 8.5 Hz), 2.46 (3H, s), 2.33 (3H, s), 2.31 (3H, s); ^13^C{^1^H} NMR (100 MHz, CDCl_3_) *δ*: 149.3, 146.3, 141.2, 139.4, 138.5, 135.5, 132.5, 132.4, 131.8, 130.7, 128.9, 128.8, 123.8, 122.8, 120.0, 118.9, 93.5, 86.6, 53.5, 21.4, 20.8, 19.6; MS (ESI-TOF) *m*/*z*: [M + Na]^+^ 483; HRMS (ESI-TOF) *m*/*z*: [M + Na]^+^ Calcd for C_26_H_24_N_2_NaO_4_S 483.1355; Found 483.1346.

*N*-Allyl-*N*-(2,4-dimethyl-6-(phenylethynyl)phenyl)-4-nitrobenzenesulfonamide (**2p**). In accordance with the experimental procedure for the synthesis of **2c**, **2p** was prepared from **1p** (122 mg, 0.3 mmol). The reaction was conducted for 7 h at −20 °C. Purification of the residue by column chromatography (hexane/AcOEt = 20) gave **2p** (132 mg, 98%). The ee (92% ee) of **2p** was determined by HPLC analysis using a chiral column (CHIRALCEL OD-3) (25 cm × 0.46 cm i.d.; 15% *i*-PrOH in hexane; flow rate, 0.8 mL/min; (+)-**2p** (major); *t*_R_ = 13.1 min, (-)-**2p** (minor); *t*_R_ = 9.9 min). **2p**: yellow oil; IR (neat) 1520, 1344, 1165 cm^−1^; [α]_D_ = +193.1° (90% ee, CHCl_3_, c = 1.07); ^1^H NMR (400 MHz, CDCl_3_) *δ*: 7.98 (2H, d, *J* = 8.5 Hz), 7.93 (2H, d, *J* = 8.5 Hz), 7.27–7.29 (1H, m), 7.21 (2H, t, *J* = 7.6 Hz), 7.13 (1H, s), 7.12 (1H, s), 6.98 (2H, d, *J* = 8.5 Hz), 5.98 (1H, dddd, *J* = 16.6, 9.5, 7.1, 5.2 Hz), 5.14 (1H, d, *J* = 17.1 Hz), 5.11 (1H, d, *J* = 10.0 Hz), 4.56 (1H, dd, *J* = 14.2, 5.7 Hz), 4.28 (1H, dd, *J* = 14.2, 8.5 Hz), 2.46 (3H, s), 2.32 (3H, s); ^13^C{^1^H} NMR (100 MHz, CDCl_3_) *δ*: 149.2, 146.3, 141.2, 138.5, 135.4, 132.7, 132.3, 131.9, 130.7, 128.9, 128.8, 128.2, 123.8, 122.6, 121.9, 119.9, 93.2, 87.1, 53.5, 20.7, 19.5; MS (ESI-TOF) *m*/*z*: [M + Na]^+^ 469; HRMS (ESI-TOF) *m*/*z*: [M + Na]^+^ Calcd for C_25_H_22_N_2_NaO_4_S 469.1198; Found 469.1170.

*N*-Allyl-*N*-(2,4-dimethyl-6-(phenylethynyl)phenyl)benzenesulfonamide (**2q**). In accordance with the experimental procedure for the synthesis of **2c**, **2q** was prepared from **1q** (108 mg, 0.3 mmol). The reaction was conducted for 7 h at −20 °C. Purification of the residue by column chromatography (hexane/AcOEt = 10) gave **2q** (127 mg, quant). The ee (86% ee) of **2q** was determined by HPLC analysis using a chiral column (CHIRALPAK AD-H) (25 cm × 0.46 cm i.d.; 15% *i*-PrOH in hexane; flow rate, 0.8 mL/min; (+)-**2q** (major); *t*_R_ = 9.1 min, (-)-**2q** (minor); *t*_R_ = 10.3 min). **2q**: yellow oil; IR (neat) 1343, 1157 cm^−1^; [α]_D_ = +172.1° (85% ee, CHCl_3_, c = 0.97); ^1^H NMR (400 MHz, CDCl_3_) *δ*: 7.84–7.86 (2H, m), 7.22–7.31 (6H, m), 7.17 (1H, d, *J* = 1.9 Hz), 7.07–7.10 (3H, m), 6.00 (1H, dddd, *J* = 17.1, 10.4, 8.5, 5.7 Hz), 5.09 (1H, dd, *J* = 17.1, 1.4 Hz), 5.05 (1H, d, *J* = 10.4 Hz), 4.51 (1H, ddt, *J* = 14.2, 5.7, 1.4 Hz), 4.29 (1H, dd, *J* = 14.2, 8.5 Hz), 2.43 (3H, s), 2.31 (3H, s); ^13^C{^1^H} NMR (100 MHz, CDCl_3_) *δ*: 141.0, 140.7, 137.9, 136.0, 133.0, 132.4, 132.2, 131.8, 131.3, 128.6, 128.2, 127.9, 127.7, 123.3, 122.6, 119.1, 93.1, 87.3, 53.2, 20.7, 19.5; MS (ESI-TOF) *m*/*z*: [M + Na]^+^ 424; HRMS (ESI-TOF) *m*/*z*: [M + Na]^+^ Calcd for C_25_H_23_NNaO_2_S 424.1347; Found 424.1347.

*N*-Allyl-*N*-(2,4-dimethyl-6-(*p*-tolylethynyl)phenyl)methanesulfonamide (**2r**). In accordance with the experimental procedure for the synthesis of **2c**, **2r** was prepared from **1r** (94 mg, 0.3 mmol). The reaction was conducted for 6 h at −20 °C. Purification of the residue by column chromatography (hexane/AcOEt = 20) gave **2r** (83 mg, 78%). The ee (87% ee) of **2r** was determined by HPLC analysis using a chiral column (CHIRALPACK AS-H) (25 cm × 0.46 cm i.d.; 5% *i*-PrOH in hexane; flow rate, 0.8 mL/min; (+)-**2r** (major); *t*_R_ = 32.9 min, (-)-**2r** (minor); *t*_R_ = 27.9 min). **2r**: yellow oil; IR (neat) 2207, 1335, 1150 cm^−1^; [α]_D_ = +7.7° (84% ee, CHCl_3_, c = 0.57); ^1^H NMR (400 MHz, CDCl_3_) *δ*: 7.39 (2H, d, *J* = 8.1 Hz), 7.25 (1H, d, *J* = 1.4 Hz), 7.19 (2H, d, *J* = 8.1 Hz), 7.08 (1H, d, *J* = 1.4 Hz), 6.00 (1H, dddd, *J* = 17.1, 10.0, 8.1, 5.7 Hz), 5.14 (1H, dd, *J* = 17.1, 1.0 Hz), 5.09 (1H, d, *J* = 10.0 Hz), 4.40 (1H, dd, *J* = 14.2, 5.7 Hz), 4.34 (1H, dd, *J* = 14.2, 8.1 Hz), 3.13 (3H, s), 2.39 (6H, s), 2.31 (3H, s); ^13^C{^1^H} NMR (100 MHz, CDCl_3_) *δ*: 140.9, 139.1, 138.2, 136.2, 133.1, 132.5, 131.8, 131.1, 129.4, 123.1, 119.5, 119.3, 93.8, 87.3, 53.7, 41.0, 21.5, 20.8, 19.4; MS (ESI-TOF) *m*/*z*: [M + Na]^+^ 376; HRMS (ESI-TOF) *m*/*z*: [M + Na]^+^ Calcd for C_21_H_23_NNaO_2_S 376.1347; Found 376.1342.

*N*-Allyl-*N*-(2,4-dimethyl-6-(*p*-tolylethynyl)phenyl)-2,4,6-trimethylbenzenesulfonamide (**2s**). In accordance with the experimental procedure for the synthesis of **2c**, **2s** was prepared from **1s** (125 mg, 0.3 mmol). The reaction was conducted for 23 h at −20 °C. Purification of the residue by column chromatography (hexane/AcOEt = 30) gave **2s** (117 mg, 85%). The ee (63% ee) of **2s** was determined by HPLC analysis using a chiral column (CHIRALCEL OD-3) (25 cm × 0.46 cm i.d.; 1% *i*-PrOH in hexane; flow rate, 0.8 mL/min; (+)-**2s** (major); *t*_R_ = 13.2 min, (-)-**2s** (minor); *t*_R_ = 10.1 min). **2s**: white solid; mp 171–172 °C (62% ee); IR (neat) 2212, 1337, 1155 cm^−1^; [α]_D_ = +155.4° (62% ee, CHCl_3_, c = 1.00); ^1^H NMR (400 MHz, CDCl_3_) *δ*: 7.14 (1H, d, *J* = 2.4 Hz), 7.05–7.08 (3H, m), 7.02 (2H, d, *J* = 8.5 Hz), 6.73 (2H, s), 6.00 (1H, dddd, *J* = 17.1, 10.0, 8.5, 5.7 Hz), 5.07 (1H, dd, *J* = 17.1, 1.4 Hz), 5.04 (1H, d, *J* = 10.0 Hz), 4.63 (1H, ddt, *J* = 14.2, 5.7, 1.4 Hz), 4.42 (1H, dd, *J* = 14.2, 8.5 Hz), 2.44 (6H, s), 2.40 (3H, s), 2.36 (3H, s), 2.29 (3H, s), 2.09 (3H, s); ^13^C{^1^H} NMR (100 MHz, CDCl_3_) *δ*: 141.5, 141.1, 139.5, 138.3, 137.7, 136.0, 135.8, 133.3, 132.2, 132.1, 131.6, 131.1, 128.7, 124.4, 119.9, 119.0, 93.2, 86.6, 53.1, 24.2, 21.5, 20.8, 20.7, 19.6; MS (ESI-TOF) *m*/*z*: [M + Na]^+^ 480; HRMS (ESI-TOF) *m*/*z*: [M + Na]^+^ Calcd for C_29_H_31_NNaO_2_S 480.1973; Found 480.1960.

*N*-(2-(*p*-Tolylethynyl)phenyl)-4-methylbenzenesulfonamide (**6**). In accordance with the experimental procedure for the synthesis of **1b**, **6** was prepared from 2-(4-tolylethynyl)-4methylaniline (558 mg, 2.7 mmol, commercially available) and 4-tosyl chloride (567 mg, 3.0 mmol). The reaction was conducted for 1 h at 0 °C–rt. Purification of the residue by column chromatography (hexane/AcOEt = 20) gave **6** (835 mg, 86%). **6**: white solid; mp 126–128 °C; IR (neat) 3239, 2212, 1335, 1159 cm^−1^; ^1^H NMR (400 MHz, CDCl_3_) *δ*: 7.68 (2H, dd, *J* = 6.6, 1.9 Hz), 7.63 (1H, dd, *J* = 8.5, 0.9 Hz), 7.35–7.38 (3H, m), 7.24–7.30 (2H, m), 7.20 (2H, d, *J* = 7.6 Hz), 7.17 (2H, d, *J* = 8.1 Hz), 7.06 (1H, td, *J* = 7.6, 0.9 Hz), 2.40 (3H, s), 2.33 (3H, s); ^13^C{^1^H} NMR (100 MHz, CDCl_3_) *δ*: 144.0, 139.3, 137.3, 135.9, 131.8, 131.4, 129.5, 129.4, 129.3, 127.2, 124.5, 120.2, 118.8, 114.8, 96.3, 83.0, 21.6, 21.5; MS (ESI-TOF) *m*/*z*: [M + Na]^+^ 384; HRMS (ESI-TOF) *m*/*z*: [M + Na]^+^ Calcd for C_22_H_19_NNaO_2_S 384.1034; Found 384.1063.

*N*-Allyl-*N*-(2-(*p*-tolylethynyl)phenyl)-4-methylbenzenesulfonamide (**3**). In accordance with the experimental procedure for the synthesis of **2c**, **3** was prepared from **6** (109 mg, 0.3 mmol). The reaction was conducted for 20 h at −20 °C. Purification of the residue by column chromatography (hexane/AcOEt = 20) gave **3** (150 mg, quant). **3**: white solid; mp 66–69 °C; IR (neat) 2214, 1344, 1159 cm^−1^; ^1^H NMR (400 MHz, CDCl_3_) *δ*: 7.64 (2H, d, *J* = 8.5 Hz), 7.48 (1H, m), 7.27–7.34 (3H, m), 7.09–7.17 (6H, m), 5.87 (1H, ddt, *J* = 17.1, 10.0, 6.6 Hz), 5.10 (1H, dd, *J* = 17.1, 0.9 Hz), 5.05 (1H, dd, *J* = 10.0, 0.9 Hz), 4.38 (2H, d, *J* = 6.6 Hz), 2.38 (3H, s), 2.23 (3H, s); ^13^C{^1^H} NMR (100 MHz, CDCl_3_) *δ*: 143.1, 139.6, 138.7, 137.1, 133.28, 133.26, 132.4, 131.3, 129.4, 128.9, 128.6, 128.0, 127.6, 123.8, 119.7, 118.6, 94.4, 85.7, 53.1, 21.5, 21.3; MS (ESI-TOF) *m*/*z*: [M + Na]^+^ 424; HRMS (ESI-TOF) m/z: [M + Na]^+^ Calcd for C_25_H_23_NNaO_2_S 424.1347; Found 424.1374.

*N*,*N-*Bis(2-bromo-4-methyl-6-(4-tolylethynyl)phenyl)benzene-1,3-disulfonamide (**4**). In accordance with the experimental procedure for the synthesis of **1b**, **4** was prepared from 2-bromo-4methyl-6-(4-tolylethynyl)aniline (1.694 g, 5.6 mmol) and benzene-1,3-disulfonyl chloride (704 mg, 2.6 mmol). Purification of the residue by column chromatography (hexane/AcOEt = 15 and then 5) gave **4** (266 mg, 13%). **4**: yellow solid; mp 268−270 °C; IR (neat) 3246, 2211, 1346, 1161 cm^−1^; ^1^H NMR (400 MHz, (CD_3_)_2_SO) *δ*: 10.23 (2H, s), 8.07 (1H, s), 7.84 (2H, d, *J* = 7.3 Hz), 7.44–7.48 (3H, m), 7.38 (2H, s), 7.30 (4H, d, *J* = 7.9 Hz), 7.20 (4H, d, *J* = 7.9 Hz), 2.32 (6H, s), 2.27 (6H, s); ^13^C{^1^H} NMR (100 MHz, (CD_3_)_2_SO) *δ*: 143.2, 139.6, 138.9, 133.8, 132.9, 132.5, 131.5, 130.3, 130.1, 129.2, 126.0, 124.7, 124.6, 119.1, 94.0, 86.2, 21.2, 19.9; MS (ESI-TOF) *m*/*z*: [M + Na]^+^ 827; HRMS (ESI-TOF) *m*/*z*: [M + Na]^+^ Calcd for C_38_H_30_Br_2_N_2_NaO_4_S_2_ 826.98705; Found: 826.98759.

*N*,*N*-Diallyl-*N*,*N*-bis(2-bromo-4-methyl-6-(4-tolylethynyl)phenyl)benzene-1,3-disulfonamide (*chiral*-**5** and *meso*-**5**). Under N_2_ atmosphere, to **4** (241 mg, 0.3 mmol) in THF (2.5 mL) was added NaH (60% assay, 24 mg, 0.6 mmol) at 0 °C, and the mixture was stirred for 20 min at −20 °C. (Allyl-Pd-Cl)_2_ (4.8 mg, 0.044 mmol), (*S*,*S*)-Trost ligand (21 mg, 0.088 mmol) and allyl acetate (194 μL, 1.8 mmol) in THF (1.5 mL) were added to the reaction mixture, and then the mixture was stirred for 21 h at −20 °C. The mixture was poured into 1N HCl solution and extracted with AcOEt. The AcOEt extracts were washed with brine, dried over MgSO_4_, and evaporated to dryness. Purification of the residue by column chromatography (hexane/AcOEt = 10) gave the mixture of *chiral*-**5** and *meso*-**5** (232 mg, 88%). The ratio (3.1:1) of *chiral*-**5** and *meso*-**5** was determined by ^1^H NMR. MPLC of the mixture gave *chiral*-**5** (147 mg, less polar) and *meso*-**5** (45 mg, more polar). The ee (99% ee) of *chiral*-**5** was determined by HPLC analysis using a chiral column (CHIRALPAK AD-H) (25 cm × 0.46 cm i.d.; 10% *i*-PrOH in hexane; flow rate, 0.8 mL/min; (–)-*chiral*-**5** (major); *t*_R_ = 14.1 min, (+)-*chiral*-**5** (minor); *t*_R_ = 17.0 min). *chiral*-**5:** white solid; mp 93–95 °C (99% ee), 89−94 °C (racemate); [α]_D_^25^ = −18.8 (99% ee, CHCl_3,_ c 1.00); IR (neat) 2212, 1354, 1088 cm^−1^; ^1^H NMR (400 MHz, CDCl_3_) *δ*: 8.44 (1H, t, *J* = 1.8 Hz), 7.78 (2H, dd, *J* = 7.3, 1.2 Hz), 7.40 (2H, d, *J* = 1.2 Hz), 7.28 (2H, d, *J* = 1.8 Hz), 7.06–7.20 (9H, m), 6.00 (2H, ddt, *J* = 17.1, 9.8, 7.0 Hz), 5.08 (2H, dd, *J* = 17.1, 1.2 Hz), 5.03 (2H, dd, *J* = 9.8, 1.2 Hz), 4.36 (2H, dd, *J* = 14.0, 7.3 Hz), 4.30 (2H, dd, *J* = 15.9, 6.7 Hz), 2.34 (6H, s), 2.31 (6H, s); ^13^C{^1^H} NMR (100 MHz, CDCl_3_) *δ*: 141.9, 140.0, 139.1, 136.0, 134.2, 133.0, 132.3, 131.5, 131.4, 129.3, 129.0, 127.4, 127.0, 126.7, 119.7, 118.9, 95.0, 85.8, 53.3, 21.6, 20,6; MS (ESI-TOF) *m*/*z*: [M + Na]^+^ 907; HRMS (ESI-TOF) *m*/*z*: [M + Na]^+^ Calcd for C_44_H_38_Br_2_N_2_NaO_4_S_2_ 907.04965; Found: 907.04749. *meso*-**5**: white solid; mp 89−94 °C; IR (neat) 2211, 1348, 1159 cm^−1^; ^1^H NMR (400 MHz, CDCl_3_) *δ*: 8.38 (1H, s), 7.76 (2H, dd, *J* = 7.9, 1.8 Hz), 7.41 (2H, d, *J* = 1.2 Hz), 7.28 (2H, d, *J* = 1.8 Hz), 7.07–7.17 (9H, m), 6.01 (2H, ddt, *J* = 17.1, 9.8, 6.7 Hz), 5.08 (2H, d, *J* = 17.1 Hz), 5.03 (2H, d, *J* = 9.8 Hz), 4.34 (4H, d, *J* = 6.7 Hz), 2.34 (6H, s), 2.31 (6H, s); ^13^C{^1^H} NMR (100 MHz, CDCl_3_) *δ*: 141.8, 140.0, 139.1, 135.9, 134.2, 133.0, 132.3, 131.4, 129.3, 129.0, 127.5, 126.9, 126.8, 119.7, 118.9, 95.0, 85.7, 53.3, 21.6, 20,6; MS (ESI-TOF) *m*/*z*: [M + Na]^+^ 907; HRMS (ESI-TOF) *m*/*z*: [M + Na]^+^ Calcd for C_44_H_38_Br_2_N_2_NaO_4_S_2_ 907.04965; Found: 907.04712.

### 3.3. X-ray Single Crystal Structural Analysis

The single crystal X-ray structures were determined by a Bruker D8 Quest with Mo*K*α radiation (*λ* = 0.71073 Å) generated at 50 kV and 1 mA. The crystal was coated by paratone-N oil and measured at 100 K. The SHELXT program was used for solving the structures [42]. Refinement and further calculations were carried out using SHELXL [43]. The chiral crystal (*P*)-**2o** (CCDC 2210583) shows the correct absolute structure and the flack parameter is 0.02(6). These data can be obtained free of charge via http://www.ccdc.cam.ac.uk/conts/retrieving.html (accessed on 1 October 2022).

## 4. Conclusions

We found that the *N*-allylation of secondary sulfonamides bearing a 2-ethynyl-6-methylphenyl group on the nitrogen atom proceeds with good enantioselectivity in the presence of (*S*,*S*)-Trost ligand-(allyl-PdCl)_2_ catalyst, giving optically active N-C axially chiral *N*-allylated sulfonamides with good yields. The N-C axially chiral sulfonamide products were also revealed to possess relatively high rotational barriers and can be handled without a decrease in the ee at room temperature. Furthermore, the absolute stereochemistry of the major enantiomer was determined by X-ray single crystal structural analysis and the origin of the enantioselectivity was rationally explained on the basis of a working model by Trost. In addition, the double *N*-allylation with bis-sulfonamide substrate gave a *N*-allylated product with two N-C chiral axes in a high optical purity.

## Data Availability

Not applicable.

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
