# Peer review of "Catalytic Enantioselective Synthesis of N-C Axially Chiral N-(2,6-Disubstituted-phenyl)sulfonamides through Chiral Pd-Catalyzed N-Allylation"

_molecules, 2022, doi:10.3390/molecules27227819_

Round 1
Reviewer 1 Report
Summary:
In this manuscript Osamu Kitagawa and co-workers have reported synthesis of rotationally stable N-C axially chiral N-allylated sulfonamides through Tsuji-Trost N-allylation of secondary sulfonamides bearing a 2-arylethynyl-6-methylphenyl group using allyl acetate in the presence of an (S, S)-Trost ligand-(allyl-PdCl)2 catalyst with up to 92% ee. The authors found that the generated N-allylated sulfonamides possess relatively high rotational barriers to display stable atropoisomeric with considerably high ee at room temperature. They also claimed to have determined the absolute stereochemistry of the major atrop-enantiomer through single crystal X-ray structural analysis and tried to explain the enantioselective generation of the major isomer through a mechanistic model.
Comments:
- In recent times, stereoselective synthesis of a significant number of N-C axially chiral atrop-isomers has been reported. As mentioned by the authors, several of these synthesises, several works of the current authors, are solely focused on achieving higher N-C axially chiral selectivity through chiral N-allyl or aryl coupling using suitable chiral catalytic complexes. As mentioned in the introduction, a previous endeavour by the same authors towards N-allylation of very similar secondary aryl-sulfonamides (bulky o-tbutyl substituted) using allyl-acetate in the presence of a very similar set of Pd based chiral catalysts resulted in several N-C axially chiral N-allylated sulfonamides in good yield and enantiomeric excess. In the present case, the authors used the same reaction condition for N-allylation of N-(2-arylethynyl-6-methylphenyl) sulfonamide to obtain a more stable N-C axially chiral N-allylated sulfonamide with higher ee. The chiral Pd catalysed stereoselective N-allylation reaction of sulfonamide to obtain N-C axially chiral N-allylated sulfonamides using allylacetate is not new, and the higher stability or higher stereoselectivity of the product in the bulkier 2,6-disubstituted sulfonamide system is a very logical and expected outcome.
- The authors here claimed to have determined the absolute stereochemistry of the major atrop-enantiomer through single crystal X-ray structural analysis. However, X-ray crystal structure provide relative arrangement of various atoms and thus give relative stereochemistry of chiral compounds. Here, as newly synthesised 2o has been crystallised as a single crystal alone without any known chirality in the crystal structure, the actual absolute stereochemistry of 2o can also be different from predicted absolute stereochemistry.
- Finally, the authors have provided a plausible model for explaining the origin of enantioselectivity in N-allylation using (S, S)-Trost ligand. The explanation of predicted enantioselectivity provided by the authors is based on an imaginary model and not on experimental evidence. The possibility of getting the single crystal structure of an intermediate complex with a Trost ligand with known absolute stereochemistry (S, S)-can provide confirmatory evidence for the absolute stereochemistry of 2o, as well as for the proposed enantioselectivity model by the authors.
In summary, the results discussed in the manuscript are not substantially novel and the evidence provided is not adequate to back up the predicted absolute stereochemistry of the products and enantioselectivity model presented by the authors. Hence, I do not recommend publication of the manuscript in its present form in Molecules.
Author Response
Thank you very much for your review and meaningful comments. Our replies to your comments are as follows.
- The chiral Pd catalysed stereoselective N-allylation reaction of sulfonamide to obtain N-C axially chiral N-allylated sulfonamides using allylacetate is not new, and the higher stability or higher stereoselectivity of the product in the bulkier 2,6-disubstituted sulfonamide system is a very logical and expected outcome. In summary, the results discussed in the manuscript are not substantially novel and the evidence provided is not adequate to back up the predicted absolute stereochemistry of the products and enantioselectivity model presented by the authors. Hence, I do not recommend publication of the manuscript in its present form in Molecules.
(Reply)
As this reviewer pointed out, it may be easily predicted that sulfonamides bearing 2,6-disubstituted-phenyl group have a rotationally stable structure. On the other hand, we think that the asymmetric induction in 2,6-di-substituted sulfonanilides is more difficult than that in ortho-mono-substituted sulfonanilides. Indeed, after screening of various ortho-substituents, we found that an ortho-ethynyl group leads to an increase in the enantioselectivity.
Furthermore, a new result describing chiral Pd-catalyzed double N-allylation was added to the manuscript. It is noteworthy that a bis-sulfonamide bearing two chiral axes was obtained with a high optical purity (99% ee) through the present double N-allylation. This result and the relevant literatures were added to the text (pages 6-7) and References 37-39 (page 21), respectively. Also, Chisato Nakamura, who conducted the double allylation, was added as a co-author. In addition, the experimental procedure, spectral data, copies of NMR chart and chiral HPLC charts relevant to the double N-allylation were added to the text (pages 18-19) and Supplementary Materials (page S42-S44 and page S64). With the additional insight provided by this new result, we believe that this manuscript should now meet the criteria for publication in Molecules.
- The authors here claimed to have determined the absolute stereochemistry of the major atrop-enantiomer through single crystal X-ray structural analysis. However, X-ray crystal structure provide relative arrangement of various atoms and thus give relative stereochemistry of chiral compounds. Here, as newly synthesised 2o has been crystallised as a single crystal alone without any known chirality in the crystal structure, the actual absolute stereochemistry of 2o can also be different from predicted absolute stereochemistry.
(Reply)
We think that this reviewer’s comments are out of context here the determination of absolute stereochemistry by X-ray single crystal structural analysis has generally been accepted. The flack parameter, 0.02(6), of sulfonamide 2o is nearly zero with a small s.d. (the statistical deviation range 4u is -0.10 ~ + 0.14), clearly indicating the (P)-configuration [see ref. 33]. For a valid absolute-structure determination, one needs both a strong or an enantiopure sufficient inversion-distinguishing power and a Flack value close to zero within statistical fluctuations. Thus, for our case assures a valid absolute-structure determination which is not twinned by inversion and for which the refined atomic model and the crystal correspond to each other. To avoid misleading the reader, we added two references, 33 and 34, by Flack that objectively supports this result.
- Finally, the authors have provided a plausible model for explaining the origin of enantioselectivity in N-allylation using (S,S)-Trost ligand. The explanation of predicted enantioselectivity provided by the authors is based on an imaginary model and not on experimental evidence. The possibility of getting the single crystal structure of an intermediate complex with a Trost ligand with known absolute stereochemistry (S,S)-can provide confirmatory evidence for the absolute stereochemistry of 2o, as well as for the proposed enantioselectivity model by the authors.
(Reply)
The working model proposed by Trost has been widely used in various asymmetric allylation reactions using the Trost ligand-Pd catalyst and the origin of the enantioselectivity is rationally explained. In addition, the Trost model is supported by computational studies. Please see Reference 35 and 36. We believe that the transition state model in Figure 2 provides a reasonable explanation for the enantioselectivity observed in the present reaction. I would appreciate your understanding.
Reviewer 2 Report
The manuscript contains interesting results on preparing N-C axially chiral sulfonamides by allylation using a chiral palladium catalyst. Optimization of the reaction conditions and wide screening of the substrates (substitution in the alkynyl, aromatic, and sulfone parts) allowed to obtain good results both in terms of yield and asymmetric induction (up to 92% ee).
The experimental part is described in detail. Unfortunately, the SI was not included in the submission, although copies of the spectroscopic spectra were promised - be sure to complete them!
The reviewed manuscript deserves to be published after editorial proofreading.
Author Response
Thank you very much for your review. Our reply to your comment is as follows.
- The experimental part is described in detail. Unfortunately, the SI was not included in the submission, although copies of the spectroscopic spectra were promised - be sure to complete them! The reviewed manuscript deserves to be published after editorial proofreading.
(Reply)
We previously submitted Supplementary Materials containing NMR charts, chiral HPLC charts, check CIFs and the evaluation of the rotational barriers. We are not sure why this reviewer could not find the Supplementary Materials. Anyway, we include the revised Supplementary Materials in the present submission.
Reviewer 3 Report
The authors describe an asymmetric Pd-catalyzed N-alkylation reaction of sulfonamides containing N-substituted phenyl group. As presented in Scheme 1, similar reactions of sulfonamides with 2-substituted phenyl have been well developed by the same group, and this work used 2,6-disubstituted phenyl substrates for the same alkylation reaction. Thus, the novelty seems limited and this reviewer does not recommend its publication on Molecules.
Author Response
Thank you very much for your review. Our replies to your comments are as follows.
- The authors describe an asymmetric Pd-catalyzed N-alkylation reaction of sulfonamides containing N-substituted phenyl group. As presented in Scheme 1, similar reactions of sulfonamides with 2-substituted phenyl have been well developed by the same group, and this work used 2,6-disubstituted phenyl substrates for the same alkylation reaction. Thus, the novelty seems limited and this reviewer does not recommend its publication on Molecules.
(Reply)
As mentioned in response to Reviewer 1, since a new result describing chiral Pd-catalyzed double N-allylation was added to this manuscript, we think that this manuscript now meets the criteria for publication in Molecules.
Round 2
Reviewer 1 Report
As described earlier, several similar work have already been reported by the same authors and others. Therefore, as such the work in not novel. As suggested, experimental evidences in support of the mechanism and absolute stereochemistry of the products, could have make the manuscript interesting. The authors have not tried to address the raised concern properly. Therefore, I do not recommend the publication of the manuscript.